METHODS

# Jointly representing long-range genetic similarity and spatially heterogeneous isolation-by-distance

**Vivaswat Shastry**[1], **Marco Musiani**[2], **John Novembre**[3]*

**1** Committee on Genetics, Genomics and Systems Biology, University of Chicago, Chicago, Illinois, United States of America, **2** Department of Biological, Geological, and Environmental Sciences, University of Bologna, Bologna, Italy, **3** Department of Human Genetics, University of Chicago, Chicago, Illinois, United States of America

* jnovembre@uchicago.edu

**Data availability statement:** The wolves data set is provided as part of the FEEMS package in (https://doi.org/10.7554/eLife.61927) (and is also publicly available from the original

## Abstract

Isolation-by-distance patterns in genetic variation are a widespread feature of the geographic structure of genetic variation in many species, and many methods have been developed to illuminate such patterns in genetic data. However, long-range genetic similarities also exist, often as a result of rare or episodic long-range gene flow. Jointly characterizing patterns of isolation-by-distance and long-range genetic similarity in genetic data is an open data analysis challenge that, if resolved, could help produce more complete representations of the geographic structure of genetic data in any given species. Here, we present a computationally tractable method that identifies long-range genetic similarities in a background of spatially heterogeneous isolation-by-distance variation. The method uses a coalescent-based framework, and models long-range genetic similarity in terms of directional events with source fractions describing the fraction of ancestry at a location tracing back to a remote source. The method produces geographic maps annotated with inferred long-range edges, as well as maps of uncertainty in the geographic location of each source of long-range gene flow. We have implemented the method in a package called `FEEMSmix` (an extension to `FEEMS`), and validated its implementation using simulations representative of typical data applications. We also apply this method to two empirical data sets. In a data set of over 4,000 humans (*Homo sapiens*) across Afro-Eurasia, we recover many known signals of long-distance dispersal from recent centuries. Similarly, in a data set of over 100 gray wolves (*Canis lupus*) across North America, we identify several previously unknown long-range connections, some of which were attributable to recording errors in sampling locations. Therefore, beyond identifying genuine long-range dispersals, our approach also serves as a useful tool for quality control in spatial genetic studies.

publication, https://doi.org/10.1111/mec.13364). This data set can be found at https://doi.org/10.5061/dryad.c9b25. The corrected wolves data set and the human data set used in this study can be found at https://doi.org/10.5061/dryad.p8cz8wb18 and https://zenodo.org/records/15007585. All simulated data can be reproduced using code in https://github.com/VivaswatS/feems/tree/admixture_edge. Finally, FEEMSmix is readily available as a complete python package from https://github.com/NovembreLab/feems.

**Funding:** Funding to JN was provided by NIH NIGMS grants R35 GM149521 and R01 GM132383. MM was supported by European Union - NextGenerationEU, under the National Recovery and Resilience Plan (NRRP), Project title "National Biodiversity Future Center -NBFC" (project code CN 00000033). The funders had no role in study design, data collection and analysis, decision to publish, or preparation of the manuscript.

**Competing interests:** The authors have declared that no competing interests exist.

## Author summary

The movement of individuals across landscapes shapes genetic diversity and has significant implications for both evolutionary studies and conservation efforts. Advances in sequencing now allow researchers to analyze thousands of samples from broad geographic areas, helping to estimate local gene flow. However, long-range genetic flow can occur due to a host of reasons (e.g., natural weather patterns, migration for resources, etc.), and existing methods struggle to represent these patterns. In this study, we developed a method to identify and model these long-range genetic similarities as dispersals from a source to a destination over a landscape. In applying this method to over 4,000 human samples from Afro-Eurasia, we detected signatures of known long-distance dispersals from recent centuries. In applying this method to 100 gray wolf samples from North America, we found many unexpected long-range genetic connections, some of which turned out to be recording errors in sample locations. Thus, beyond detecting real long-range dispersal, our approach also serves as a useful tool for quality control in spatial genetic studies.

## Introduction

A key first step in understanding the genetics of a species is to understand its variation across the geographic range it inhabits (i.e., the geographic structure of genetic variation, or the "landscape genetics" of the species [1–4]). In many, or most species, isolation-by-distance patterns are common, in which genetic similarity is highest amongst the most geographically proximal individuals (i.e., [5]). The scaling of isolation-by-distance patterns often varies across a species's range ("spatially heterogeneous isolation by distance") due to factors such as persistent, prominent geographic features that alter migration, or non-equilibrium dynamics such as expansions from glacial refugia.

Models of spatially heterogeneous isolation-by-distance developed over the past decades have proven to be quite useful. In these models, local migration rates connecting neighboring local populations (or demes) are allowed to vary across the landscape. Methods based on such models can take geographic or ecological data as input and produce predictive maps of expected genetic differentiation [6,7], or they can be used to fit regression weights for ecological factors that might affect observed genetic connectivity [8]. The models can also be used with genetic data as input to produce maps of high and low levels of local gene flow [9,10], as is our focus here.

However, one limitation of models of spatially heterogeneous isolation-by-distance is their inability to model long-range genetic similarity. In many taxa, localized gene flow is punctuated by pulses of long-distance dispersal across a landscape [11]. Such events produce cases where individuals separated by long geographic distances are remarkably genetically similar to one another. Ideally, methods to represent the geographic structure of genetic variation can also help identify when such long-range genetic similarities exist.

Being able to identify the location of the sources and destinations of recent putative long-range gene flow events can not only be useful in painting a more complete picture of the recent evolutionary history of dispersal and reproduction in a species, but can also be relevant for conservation in terms of identifying long-range genetic connectivity which may affect the long-term survival of populations [12], and the functioning of ecosystems and the services ecosystems provide [13,14].

To address this challenge, we present an extension of the EEMS ("Estimation of Effective Migration Surfaces") method from [9] and FEEMS ("Fast EEMS") method from [10] that represent spatial genetic structure by inferring migration rates on the edges of a graph of connected nodes. The default graph is dense and has neighboring nodes connected to one another such that the resulting set of symmetric migration rates across the edges approximately specifies a continuous "migration surface". The nodes in the graph represent local populations (i.e. sets of randomly mating individuals) and are referred to as demes for the remainder of the manuscript. On this graph, edges with low inferred migration help convey how samples are more dissimilar than expected given their geographic separation. Conversely, regions of high inferred migration depict when samples are more similar than expected given their geographic separation. We refer interested readers to [9] and [10] for a full explanation of the model and to [15] for limitations of the model with regards to the modeling of directional/asymmetric migration .

Our extension, called FEEMSmix, detects when the nearest neighbor graph is insufficient to explain the data and adds directional long-range edges (LREs) to account for excess similarity between distant nodes in the graph. This similarity might arise from long-range gene flow or admixture events, or mistakes in record-keeping of the geographic position assigned to a sample. Long-range genetic similarity could arise due to a single recent, instantaneous pulse or from some form of continuous gene flow stretching into the past over a region of very low effective migration. In all cases, our method will represent these events in the form of an interpretable *source fraction* parameter, akin to an admixture proportion, from a hypothetical instantaneous long-range pulse that occurs just before sampling.

This framework follows closely from existing methods for characterizing population structure that model outlier relationships using an admixture component (TreeMix, [16]; MixMapper, [17]; SpaceMix, [18]). TreeMix first established the approach with an initial step of fitting a tree topology to explain the genetic similarity among populations, and then adding "admixture edges" between populations that show high residuals and refitting the tree topology. The approach stops after a user-defined $K$ number of edges are added. SpaceMix implemented a similar approach but using latent genetic space variables in a model with spatial genetic covariance and modeling each individual as potentially admixed with each other population as a source. Relative to these two prior approaches, FEEMSmix differs in several key ways. First, the baseline model used is the EEMS/FEEMS framework, such that our output provides an integrated representation of spatially varying local genetic structure and long-range events. Second, we explicitly model long-range connectivity in terms of adding specific pairwise directed edges (like TreeMix) though with specific geographic source locations to aid interpretation and give geographic context for understanding signals of long-range genetic similarity.

In this manuscript, we present the FEEMSmix method and test it with simulations over a range of parameter values to quantify its performance. Finally, we apply this method to two large empirical data sets across different species and geographic ranges: first, to a data set of 111 wolf samples across North America (originally from [19], and used as a prototype in FEEMS), and second, to a data set of 4,070 humans from over 300 sampling locations across Africa, Europe and Asia (compiled as part of [20]) to validate the working of the method in a well-studied system with findings that can be corroborated with alternative historical data sources. Like with TreeMix we encourage exploration of a set of user-defined $K$ additional admixture edges, however we also explore likelihood and $R^2$-based heuristics for evaluating the support for any given edge.

## Description of the method

### Analysis of a representative simulated dataset

We show a schematic workflow for the methodology of FEEMSmix using a representative simulation of a simple scenario of spatial population structure with a long-range gene flow event via Fig 1.

For the simple scenario, we consider a spatial graph of locally connected demes that form a $8 \times 12$ grid with a barrier region of low migration running "North" to "South" across the center of the grid. The effective migration rates inside the barrier are ten times lower than the adjacent areas, resulting in an $F_{ST} \approx 0.1$ across the barrier. The scenario includes an instantaneous long-range event going "West" (source) to "East" (destination) across this barrier just prior to sampling, with a backward migration fraction of $c = 0.5$. In other words, looking backward in time, 50% of the lineages in the destination deme migrate in a pulse-like fashion to the source deme. In FEEMSmix, we call this parameter the *source fraction* as it is always associated with a particular source and it represents the fraction of lineages that belong to this particular source going backward in time. To demonstrate the performance under two extremes of data availability, we use two sampling schemes called *constant, dense sampling* and *variable, sparse sampling* (shown in S1 Fig). In the former scheme, we impose a 50% sparsity and sample the true source deme, with 10 individuals per sampled deme across the grid. In the latter scheme, we impose a strict sparse sampling to mimic real-world settings. More than 80% of the demes in the graph are unsampled, and we leave the true source deme of the long-range gene flow event also unsampled. In this case, we sample a random number of individuals between 1 and 10 per sampled deme. We note that in real data applications from continuous landscapes, users must choose the resolution of the grid to use for analysis, and helper functions in the software to help register their samples onto the grid. The full simulation parameters can be found under *Simulation settings* in Verification and comparison.

**Step 1. Apply FEEMS.** We fit the FEEMS method to the data using the deme-specific variance mode outlined in [10], with cross-validation to choose the appropriate tuning parameters $\lambda$ and $\lambda_q$. We refer to the resulting fit as the 'baseline' fit in the remainder of the paper.

In the example in Fig 1, we see that FEEMS does a mediocre job of reconstructing the low dispersal barrier region. In particular, it fits the regions poorly between the destination and source of the long-range gene flow event – as it compensates for the high genetic similarity between the source and destination demes on either side of the barrier by fitting a corridor of high gene flow between the two demes.

**Step 2. Identify candidate poorly fit pairs of demes.** In this step, given our motivation to detect and describe long-range genetic similarity that is not well represented by the FEEMS model, we identify pairs of demes that have a smaller observed (genetic) distance than expected under the FEEMS model. To do this, we calculate a deviation statistic ($x_{ij}$ for a pair of demes $\{i, j\}$) that is the centered and standardized logarithm of the ratio between the observed and fitted genetic distances. We designate the top 5% of points with the largest negative deviation as candidate outliers.

Even though we see a reasonably high $R^2 \approx 0.82$ in this representative example (Fig 1, Step 2), there are pairs of demes that are not well fit under the model (indicated by the red circles, Step 2 panel).

Importantly, we see visually that the distribution of the deviation under the baseline fit can be modeled as a mixture of two univariate Normal distributions (in gray and red, see inset histogram in Step 2). Quantitatively, we evaluate the support for a two-component mixture model using a ratio of the maximum likelihood values under each model: $L_r = -2 \ln \left( \hat{L}_1 / \hat{L}_2 \right)$,

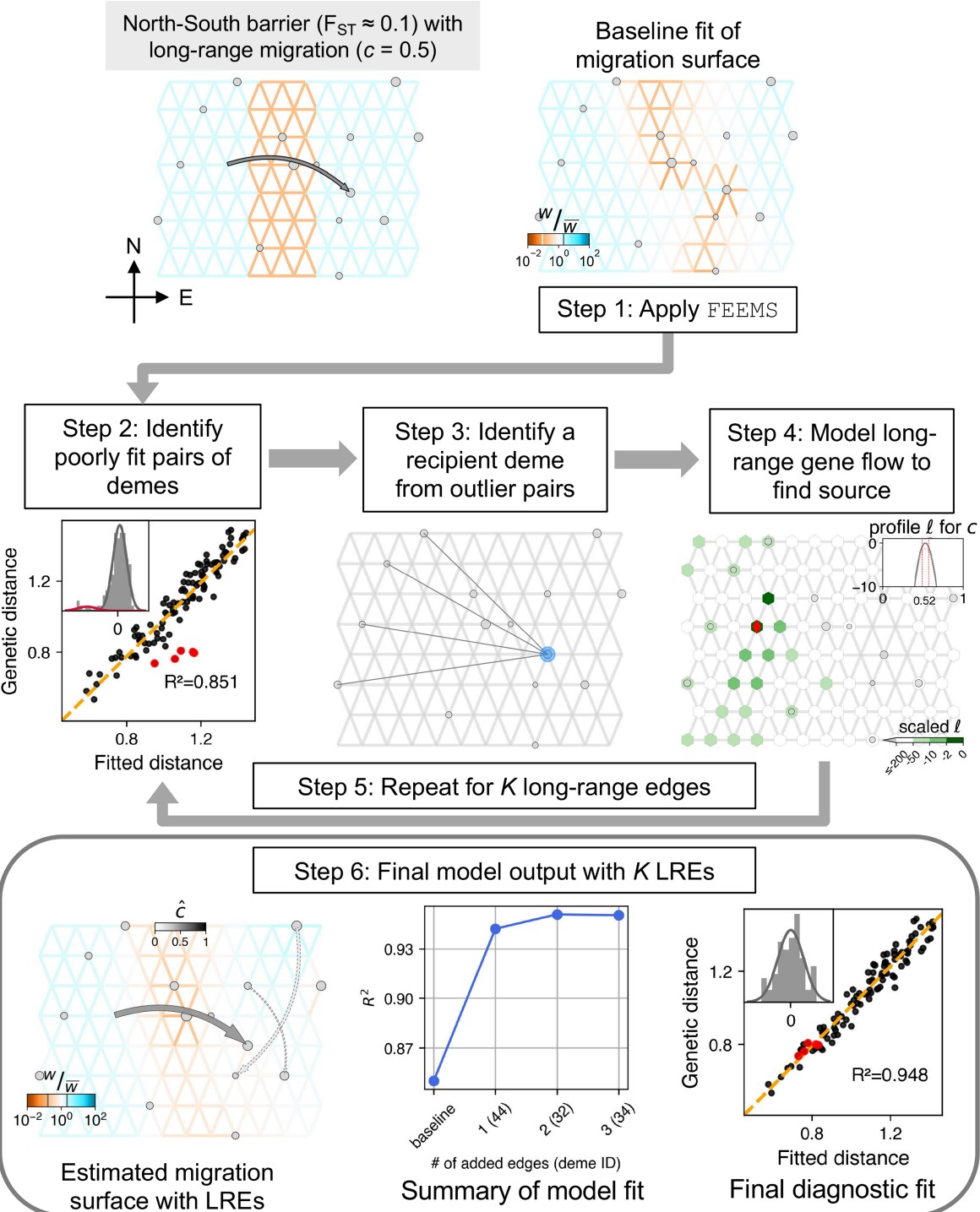

**Fig 1. Summary of workflow of the FEEMSmix method presented via application to a representative simulation of a single barrier scenario with *variable, sparse sampling* (see main text for description of the steps and S2–S3 Figs for the *constant, dense sampling* analog).** The background effective migration rate in the corridors are 10× larger with $\overline{Nm} = 0.1$ compared to the barrier region with $\overline{Nm} = 0.01$. In the simulated scenario (top left panel) and fitted results, the weights ($w$) on the edges that represent migration rates between pairs of demes are shown relative to their mean value across all edges (see legend for $w/\bar{w}$).

with $\hat{L}_1$ representing the fit of the deviations to a single Normal distribution, and $\hat{L}_2$ the fit to a two-component mixture of Normals. We take values of $L_r > 10$ to represent substantial support for two-components in the residuals and in turn support for the presence of a long-range edge. This approach is necessarily heuristic, as the use of penalized likelihood precludes more exact or asymptotic approaches to model selection.

**Step 3. Identifying a putative recipient deme from outlier deme pairs.** As a visual aid, we draw edges between the candidate outlier deme pairs identified in Step 2. As shown in this representative example, a single simulated long-range dispersal event between a source and destination deme will typically cause multiple demes neighboring the source to be fit poorly by the baseline FEEMS model. While this provides a summary of the FEEMS fit that emphasizes potential unmodeled long-range genetic similarity, we model a long-range gene flow event between a single source deme to a single recipient deme as a compact way of representing and explaining the signal observed across multiple outlier pairs. To do so, we first identify putative recipient demes from the candidate outlier pairs to help narrow our search space down from order of $\mathcal{O}(o^2)$ to $\mathcal{O}(o)$, where $o$ is the number of observed demes.

We consider each outlier edge as specifying a potential long-range gene flow event and assess the favored directionality of gene flow for each outlier edge. We use the following algorithm: for a pair of sampled demes $i$ and $j$ implicated as an outlier pair, we fit them as the result of long-range gene flow (see Eq (1)) in both possible directions, i.e., $i \rightarrow j$ or $j \rightarrow i$ with the $\rightarrow$ denoting the direction of gene flow from source to destination forward in time. If the model fit for $i \rightarrow j$ has a log-likelihood 2 units larger than $j \rightarrow i$, we take $j$ as a putative recipient deme and $i$ if the opposite is true. In cases where these quantities are within 2 units of each other, both $i$ and $j$ are added as putative recipients to our list.

Taking this approach, destination demes are often found to be the putative recipient across multiple outlier pairs (Fig 1, Step 3). To choose the destination for the first LRE, we rank the putative recipient demes by a weighted sum statistic $f_i \propto \sum_j x_{ij} e^{-x_{ij}}$ with the sum being taken over all outlier pairs involving deme $i$. This favors recipients that are implicated in multiple outliers pairs *and* have large negative residuals. The size of the blue circle for deme $i$ is proportional to the $f_i$ statistic (shown in Fig 1, Step 3). We then take the deme with the largest negative value to be the putative recipient for the LRE to be fitted. In this case, we see that the method correctly identifies the true destination deme as the putative recipient.

**Step 4. Fitting the source location for a chosen destination** For the chosen recipient deme from Step 3, we model a long-range gene flow event and find the maximum-likelihood estimate (MLE) of the source location and corresponding genetic ancestry proportion derived from that source (i.e., source fraction $\hat{c}$ in Fig 1, Step 4 panel, also see *Specifying the FEEMSmix likelihood*). We also output several visual summaries relevant to the fitting of the source location for a single LRE: a *marginal likelihood surface* over the entire grid for the putative source location of the fitted LRE with darker green reflecting a higher log-likelihood of a particular deme being the source; an *arrow connecting from each MLE source to its corresponding destination deme* colored by $\hat{c}$ (gray-scale from white to black for [0,1]); a *profile log-likelihood curve* in gray for the estimated source fraction with dashed red lines indicating 2 log-likelihood units around the MLE and lighter grey lines in the background represent profile likelihoods for other potential sources that lie within this threshold.

For the example shown in Fig 1, the method's first LRE identifies the true recipient deme and finds the estimated source deme to be the true source (shown as a red diamond) with a reasonably accurate estimate of the source fraction ($\hat{c} = 0.52$ compared to $c = 0.5$ simulated).

**Steps 5 - 6: Additional edges and final model output.** After the fitting of the first LRE to the putative recipient deme, Steps 2 through 4 are repeated sequentially on a re-estimated migration surface that *contains* preceding LREs for a user-specified number of edges, $K$ (see

*Joint optimization of the likelihood* for details). For each edge added, the $L_r$ statistic is calculated on the deviation to measure the support for the particular edge.

The final output is a map that shows long-range arrows indicating source and destination for each LRE, colored by the associated MLE source fraction ($\hat{c}$), placed over the underlying grid with edges colored by their estimated rate parameters from a joint fit (Fig 1, Step 6). The size of the arrow decreases with each added LRE in order to visually highlight the demes with the largest residuals. Akin to the practice for adding migration edges in `TreeMix` [16], we do not impose a strict stopping criterion, though users may find it helpful to inspect various outputs when interpreting their results and evaluating the number of edges to include in the final output. To aid in this interpretation, we produce statistical summaries of each LRE, both visual and numerical:

- LREs are displayed as solid arrows if $L_r > 10$ or as dashed arrows if they do not (*Estimate migration surface with LREs* in Fig 1, Step 6). The expectation is that the solid LREs capture significant long-range signals and reduce residual variance, while dashed LREs beyond this point have lower, but not negligible, support (still implicated in top 5% of outlier pairs but with $L_r \leq 10$).
- Improvement in fit to the observed genetic distances via $R^2$ as a function of $K$ (*Summary of model fit* in Fig 1, Step 6). It is of additional interest to note the smallest value of $K$ for which there is a plateauing of $R^2$.

The method finally also produces a scatter plot of the observed genetic distance against fitted distance *after* modeling outliers *and* fitting the background `FEEMS` parameters to show the improvement in fit via $R^2$ (*Final diagnostic fit* in Fig 1, Step 6). In the inset, we also observe that the deviation statistic can be modeled with a single Normal distribution, indicating that any additional LREs on this surface will show negligible support. This corresponds to our expectation in this simulation since the first LRE has already modeled away the residuals from the single long-range gene-flow event.

In addition, because the long-range event accounts for the residual genetic similarity between the two sides of the barrier, the migration weight estimation on the graph improves, especially in the area between the source and destination demes. We also observe a higher $R^2$ value compared to the baseline `FEEMS`, and an improved fit of the previous outlier pairs (red points, Fig 1).

In this example, `FEEMSmix` finds only the first edge to significantly reduce variance in the deviation (shown as a solid arrow). We also show two additional LREs beyond the first edge for illustration purposes, and find that they are classified to have no support under our framework. These LREs only marginally improve the model $R^2$, reaching a plateau after $K = 1$, but still represent genuine residuals from the baseline `FEEMS` fit. We observe a similar pattern across a suite of simulation replicates (see S4 Fig).

Finally, a fundamental issue necessitates the use of heuristic measures (like the deviation statistic above) in the detection and validation of the fitted LREs. `FEEMS` employs penalized likelihoods determined by cross-validation, precluding the standard interpretations of likelihood ratio test statistics or information criterion-based model selection approaches like AIC or BIC [21]. This regularization framework makes it challenging to classify an edge as statistically significant or a 'true positive' using conventional hypothesis testing or model selection testing procedures. While the parametric bootstrap is a potential approach to move forward, the time demands of the computation involved make it impractical. Consequently, our heuristic approaches provide a practical solution for identifying potential long-range gene flow

events while acknowledging the inherent complications imposed by the underlying model structure and estimation procedures.

We now explain each step in more detail.

### Baseline FEEMS model

As the first step for our method, we fit the same model as FEEMS using the same input (genotype matrix and spatial locations for sampled individuals) and framework (assigning samples to closest nodes on a user-defined triangular grid, and estimating graph-specific parameters via penalized maximum-likelihood) with the same modeling assumptions (exchangability of individuals within a deme, symmetric migration, unlinked SNPs and multivariate normal assumption for the allele frequencies). We employ a modified version of FEEMS that allows for modeling deme-specific variance parameters compared to the default version that uses a single fixed variance across the grid. Although this version was originally presented in [10], FEEMS defaults to a faster, more parsimonious default version with a fixed variance because the deme-specific variances increases the number of parameters by $o-1$ (number of observed demes) and increases runtime, but we found the added robustness when fitting long-range gene flow events to be a worthwhile trade-off. More specifically, this entailed replacing the fixed term, $\sigma^2\mathrm{diag}\left(\mathbf{n}^{-1}\right)$, by a vector, $\mathbf{q}\in\mathbb{R}^o$, in the FEEMS likelihood (Eq 7 in [10]) to parameterize deme-specific variances. Also, to avoid over-fitting, a penalty term is added to the likelihood, with scalar $\lambda_q$ such that larger values impose greater similarity across the elements of $\mathbf{q}$ (see [10], for a complete description). Finally, we note that the parameters estimated in FEEMS are the migration weights $\mathbf{W}$ on the grid and the deme-specific variances $\mathbf{q}$.

### Modeling long-range genetic similarity as instantaneous gene flow events

We model gene flow along each LRE as a uni-directional instantaneous pulse event in the most recent generation such that a fraction $c\in[0,1]$ of genetic lineages in destination deme $d$ descend from a source deme $s$. Within this model, the expected pairwise coalescent time *post*-event ($T'_{ij}$) between any two demes $i$ and $j$ can be derived as a function of the expected pairwise coalescent times *pre*-event ($T_{ij}$) and this source fraction $c$:

$$\begin{aligned}
T'_{ss} &= T_{ss},\\
T'_{sd} &= cT_{ss} + (1-c)T_{sd},\\
T'_{kd} &= cT_{sk} + (1-c)T_{kd}\ \ \forall\, k\in\{1,\dots,o\}\setminus\{s,d\},\\
T'_{dd} &= (1-c)^2 T_{dd} + 2c(1-c)T_{sd} + c^2 T_{ss}.
\end{aligned} \tag{1}$$

These equations are derived by noting how the expected pairwise coalescent times are naturally mixtures over the conditional expectations that arise by considering the different possible scenarios for the ancestry of each lineage in the pair. For example, the *post*-event $T'_{sd}$ is a weighted average of $T_{ss}$ when lineage $d$ derives from $s$ (an event with probability $c$) and $T_{sd}$ when the lineage does not derive from $s$ (with probability $1-c$).

### Specifying the FEEMSmix likelihood

The software FEEMSmix is built completely on top of the FEEMS model developed by [10], but we use a parametrization of the model in terms of pairwise coalescent times developed in EEMS by [9]. This is because each approach has unique advantages: 1) the FEEMS framework provides fast gradient-based optimization machinery for penalized likelihood-based

optimization (whereas EEMS uses Markov chain Monte Carlo in a Bayesian framework), and 2) the EEMS likelihood parameterizes the genetic distance between samples in terms of pairwise coalescent times, which are more readily adapted for extending the model to more complex scenarios of gene flow.

To achieve this, we briefly connect the FEEMS likelihood to the equivalent EEMS likelihood (Eq 3 in [9], or Eq (7) in this text), and restate the former in terms of expected pairwise coalescent times between lineages sampled from any pairs of demes $i$ and $j$. First, we follow EEMS and define the expected symmetric unscaled genetic distance matrix $\mathbf{\Delta} \in \mathbb{R}^{o \times o}$ with $\text{diag}(\mathbf{\Delta}) = 0$ as a function of coalescent times that are approximated using a resistance distance matrix $\mathbf{R}$ (with $\text{diag}(\mathbf{R}) = 0$) and node-specific variance parameters $\mathbf{q}$ as follows (for explanations of the approximations see the Supplement of [9]):

$$\Delta_{ij} \approx 4T_{ij} - T_{ii} - T_{jj} \tag{2}$$

$$T_{ij} \approx R_{ij}/4 + (q_i + q_j)/2 \tag{3}$$

$$T_{ii} \approx q_i \tag{4}$$

$$\implies \Delta_{ij} \approx R_{ij} + q_i + q_j, \quad i \neq j \tag{5}$$

Importantly, the resistance distance between the two demes is given by $R_{ij} = -2L_{ij}^{\dagger} + L_{ii}^{\dagger} + L_{jj}^{\dagger}$, wherein $\mathbf{L}^{\dagger}$ is the pseudo-inverse of the graph Laplacian given by $\mathbf{L} = \text{diag}(\mathbf{W1}) - \mathbf{W}$ [6,8,10]. The pairwise coalescent time between two demes $i$ and $j$ can be decomposed into two components: the time taken for two lineages to meet in an intermediate deme (also known as the commute time and is approximately one-fourth the resistance distance $R_{ij}$ under symmetric migration), and the time for coalescence within that deme (valid under conservative migration, approximated as the average within-deme coalescent times as per [22]).

In this way, we can see how the edge weights $\mathbf{W}$ and node variances $\mathbf{q}$ estimated in FEEMS are connected to the quantities in the expected genetic distance matrix in EEMS. Based on the derivations above for the coalescent times after a long-range gene flow event from deme $s$ to deme $d$ from Eqs (1)–(5), we can formulate expressions for any off-diagonal element of the expected genetic distance matrix *post*-event $\mathbf{\Delta}'$ as a function of the source fraction $c$,

$$\Delta_{ij}' = \begin{cases} \frac{1}{2}(1-c)(2-c)R_{sd} + (1+c)q_s + (1-c)q_d, & i \in \{s\}, j \in \{d\}, \\ (1-c)R_{id} + cR_{is} - \frac{1}{2}c(1-c)R_{sd} + q_i + cq_s + (1-c)q_d, & i \notin \{s,d\}, j \in \{d\}, \\ R_{ij} + q_i + q_j, & \text{otherwise.} \end{cases} \tag{6}$$

Here, an interpolated $\hat{q}_s$ (see *Approximation of pairwise coalescent times in an unsampled deme*) will be substituted in place of $q_s$ when the source deme $s$ is unsampled. Note that for $c = 0$, these equations reduce to the baseline model given in Eq (5). When fitting $c$, we do not apply any penalty for sparsity, i.e., $\hat{c}$ is the value of $c$ that maximizes the marginal likelihood for $c$. Mathematically speaking, this is just a low-rank update (of rank 1) to the *pre*-event genetic distance matrix as a function of $c$. The intuition for the elements of the genetic distance matrix are not as straightforward as the pairwise coalescent times in Eq (1), but can be viewed as

perturbations from the baseline genetic distances in Eq (5). For instance, with $c = 1$ and with $i \in \{s\}, j \in \{d\}$, we see that the genetic distance is just $\Delta_{sd}' = 2q_s$. This makes sense as *all* the lineages in deme $d$ are from $s$, so $\Delta_{sd}'$ will not depend on the resistance distance between the two demes and is completely driven by the expected time to coalescence within deme $s$. Similarly, when $c = 1$ and $i \notin \{s,d\}, j \in \{d\}$, then the genetic distance $\Delta_{id}' = R_{is} + q_i + q_s$. Here, we

notice that there is no contribution from $R_{id}$ or $R_{sd}$ as there are *no* original lineages remaining in the destination deme $d$, effectively negating the contribution of any routes to this deme.

Using the expression for the expected pairwise distances ($\mathbf{\Delta'}$), we then follow EEMS and model the *observed* pairwise genetic distance matrix $\hat{\mathbf{D}}$ as a draw from a Wishart distribution:

$$-C\hat{\mathbf{D}}C^{\top} \sim \mathcal{W}_{o-1}\left(-\frac{\sigma^{*}}{p}C\mathbf{\Delta'}C^{\top}, p\right). \tag{7}$$

The matrix $C$ is a contrast matrix to remove the overall mean. The scaling parameter, $\sigma^{\star}$, is set to 1, as forcing it to be 1 will scale the units of $\mathbf{R}$ and $\mathbf{q}$ to be in units of expected distance. Finally, $p$ is the degrees of freedom which is set to the number of unlinked SNPs as in [9]. Consequently, we strongly recommend using only unlinked SNPs, as including linked SNPs may result in overly confident likelihood estimates. This issue is particularly salient in FEEMSmix, where the likelihoods are used to identify putative source demes across the grid.

## Approximation of pairwise coalescent times in an unsampled deme

To model the expected pairwise coalescent times *post*-event in Eq (1) when the source is an unsampled node (say, $s$) in the framework, we need an expression for the expected coalescent time within this deme, $T_{ss}$, but the baseline FEEMS model only estimates this deme-specific variance parameter for sampled demes on the grid. As a result, we provide an approximation for the coalescent time $T$ within an unsampled deme by spatially interpolating between node-specific variances at sampled demes on the grid using a kriging-like approach with an exponential variogram model defined by $q(R) = b + C_0\left(1 - \exp(-R/a)\right)$, where $b$ is the nugget, $C_0$ is the sill and $a$ is the range parameter [23,24]. Notably, here, we use the resistance distances $R$ instead of geographic distances to account for the effect of spatially heterogeneous isolation-by-distance. These three kriging parameters are estimated by fitting the variogram to inferred variances from sampled demes across the grid by minimizing the squared difference between the expected and inferred values. The weights $\gamma_{1,\cdots,o}$ for each of the observed demes are determined by solving the set of linear equations given by $\hat{q}_s = \sum_{i=1}^{o} \gamma_i q_i$ using weighted least squares optimization for a specific unsampled deme, which can then be used to formulate $\Delta'_{ij}$.

## Joint optimization of the likelihood

**With a single long-range edge.** Here, we outline the two-step procedure followed to fit a single LRE during Step 4 in Results. In our first 'pre-fitting' step, we fit a model of instantaneous gene flow from *every* deme in the grid to the putative recipient deme in an independent fashion. This is done by minimizing the negative log-likelihood of the data in Eq (7) for the source fraction $c \in [0,1]$, *while* holding the other parameters constant at their baseline values. This first step is very fast as for each source, we only optimize over a single dimension. For our second 're-fitting' step, we choose only the top fraction (e.g., 1%) of demes with the highest log-likelihoods from the previous step to perform the joint fitting procedure in which we estimate *all* parameters in the model. This is an attractive approach as it saves the effort of having to estimate joint fits for demes that have a very low likelihood of being the true source deme (i.e., a vast majority of the demes in the grid), but still provides a way of searching over the entire grid.

We employ a coordinate-descent approach when minimizing the negative log-likelihood of the model with *all* parameters for a particular LRE: edge weights, node variances *and* source fraction. Similar to the 'pre-fitting' step, we first optimize over the single dimension of the

source fraction $c$ using Eq (7) but with the parameters held constant at their values estimated from the baseline fit in FEEMS. Then, holding $c$ constant at its MLE value, we re-fit the edge weights and deme-specific variances using the likelihood from FEEMS. Then, we repeat the optimization procedure for the source fraction holding the other parameters constant at their MLE values, and so on, until an absolute tolerance is reached in the values of the parameters ($10^{-3}$ for the source fraction and $10^{-7}$ for the edge weights and deme-specific variances). We initialize the parameters at the baseline FEEMS fit to speed up convergence as we only expect slight deviations in estimates for any single LRE.

This two-step approach provides us with a fast and simple way to optimizing the joint likelihood as it allows us to reuse the fast and flexible machinery formulated in [10] for computing the gradients as a function of the edge weights and deme-specific variances. The only substitution we perform is to replace the expected covariance matrix $\mathbf{\Sigma}$ in Eq 18 of [10] with our reformulated expected covariance matrix $\mathbf{\Sigma}'$ that is simply derived from the expected distance matrix $\mathbf{\Delta}', \mathbf{\Sigma}' = -\frac{1}{2}J(\Delta'_{ij})^2 J$ (where $J = I - \frac{1}{o}\mathbf{1}\mathbf{1}^\top$ is the centering matrix and $I$ is the identity matrix [8,25]). Optimization in `FEEMSmix` is done using the Nelder-Mead and L-BFGS-B algorithms implemented in `scipy` [26–28], with the mixture model fit using `scikit-learn` [29].

**Multiple edges.** By default, `FEEMSmix` fits multiple edges using an *iterative* approach (following from `TreeMix`). First, we fit a long-range edge to the putative recipient deme with the largest number of outlier pairs from the baseline fit using the workflow in Steps 1-4 of Fig 1, and then, we fit a LRE to the next putative recipient deme over a surface fitted *containing* the previous LRE, and so on. We repeat this procedure until a user-specified $K$ edges are fit on the baseline graph. As a precautionary measure against overfitting, we default to only allow for a maximum of two LREs to be fit to the same recipient deme. However, this can be changed in the software version of the package using a command-line parameter. We also do not specify a default stopping criterion here similar to `TreeMix`, as it is difficult to have rigorous, stable criteria, so we recommend users interpret edges using summary statistics like the $L_r$ statistic, model $R^2$ and outputs from parallel methods.

Finally, we also provide users with the option of fitting multiple edges using an *independent* approach for each putative deme in the initial list of outliers from the baseline FEEMS fit.

## Verification and comparison

### Simulation settings

All simulations were conducted in `msprime` [30]. As a set of test cases, we simulate a $8 \times 12$ nearest neighbor graph/grid of demes with a barrier at the center of the grid to capture a spatially heterogeneous migration landscape. The long-range gene flow event occurs as an instantaneous pulse (`MassMigration` event) with varying source fraction $c \in \{0.05, 0.25, 0.5\}$ across the barrier. We use 1,000 independent SNPs and show results for two sampling scenarios:

- *Constant, dense sampling.* A sampling of 50% of all demes in the grid with the true source deme being sampled with 10 individuals per deme and a uniform population size of 1,000.
- *Variable, sparse sampling.* A random sample of 15% demes across the grid leaving the true source deme unsampled, and with a variable number of individuals per deme drawn uniformly from between 1 and 10. We also simulated unequal population sizes across the grid, drawing from a uniform distribution between 100 and 10,000.

In addition, we also tested several scenarios to assess the robustness of the method to various model mis-specifications, including panmixia (S12 Fig), older admixture events (S15–S16 Figs), admixture from multiple sources (S17 Fig), non-uniform sampling (S18–S20 Figs), and long-distance gene flow to multiple neighboring locations (S21–S22 Figs, also see Discussion).

## Evaluation across multiple replicate simulations

We apply the method to fifty replicates of the simulated scenario in Fig 1 and evaluate performance in terms of: 1) the rate of finding the correct destination deme, 2) the average distance from the MLE to the true source deme, and 3) the error in the estimated source fraction ($\hat{c}$). We repeat this procedure across the two sampling schemes noted above, and for values of $c \in (0, 0.05, 0.25, 0.5)$.

In the case of no long-range gene flow ($c = 0$), we added $K = 3$ LREs and tested each for statistical support under our classification criteria. We found that 4%, $C_{95} = [1\%, 8\%]$ of individual LREs falsely showed support for long-range gene flow in the dense sampling scenario and 9%, $C_{95} = [4\%, 13\%]$ in the sparse sampling scenario ($C_{95}$ indicates a 95% confidence interval computed using the Clopper-Pearson method, [31]). These false positive rates represent LREs that incorrectly rejected the null hypothesis of no systematic structure in residuals. Additionally, we also find minimally biased estimates for $c$ across the top LRE inferred by the method in both dense and sparse sampling scenarios (bias= 0.001 $[0, 0.002]$ and 0.008 $[0.003, 0.013]$ respectively with the interval representing two-times the standard error in the estimated bias, in Fig 2C). This bias decreases by approximately five-fold if we only consider LREs with $L_r > 10$ under our method.

Across both simulated scenarios, the method identified the true destination deme in 100%, $C_{95} = [93\%, 100\%]$ of the replicates with dense sampling and high gene flow, and in 92%, $C_{95} = [81\%, 98\%]$ of replicates with sparse sampling and high gene flow ($c = 0.5$). In the cases with low gene flow ($c = 0.05$) 0%, $C_{95} = [0\%, 7\%]$ of replicates had the destination recovered correctly in the dense sampling scenario, and 12%, $C_{95} = [5\%, 24\%]$ correctly in the sparse scenario. However, the average distance from the estimated destination deme to the simulated is within 3 units on the grid in both cases. This seemingly better performance with sparser sampling is plausibly due to the fact that the sparse scenario has fewer sampled demes to choose from in the area surrounding the destination of the long-range event.

For evaluating the identification of the source, we fixed the destination deme to the top implicated destination deme and fit a paired source location for it (Step 4). In cases with high gene flow, this top destination deme was the same as the simulated destination in all simulation replicates across both sampling scenarios. However, with weak gene flow and sparse sampling, this overlap dropped, with the top implicated destination being at an average of 3.2 inter-deme distance units from the true destination. In general, the correct direction of gene flow is captured with the estimated source being on average only 2 units away from the true source (see Fig 2A). When fitting cases with $c = 0.5$, the method identified the correct source in 96%, $C_{95} = [86\%, 100\%]$ of the simulations in the dense scenario and 20%, $C_{95} = [10\%, 34\%]$ simulations in the sparse scenario. In the sparse scenario, FEEMSmix picks a neighboring deme to the true source deme in 75%, $C_{95} = [62\%, 87\%]$ of the simulations (see S4 Fig for the suite of results). The mean estimation error is visually summarized in Fig 2A. As expected, uncertainty increases with weaker gene flow and sparser sampling, though the MLE remain concentrated relative to the size of the grid. In Fig 2B, we assess accuracy using a coverage statistic, i.e., the percentage of simulations in which the true source is within a threshold $x$ log-likelihood units of the MLE source. We see that for $c = 0.5$ in the dense scenario

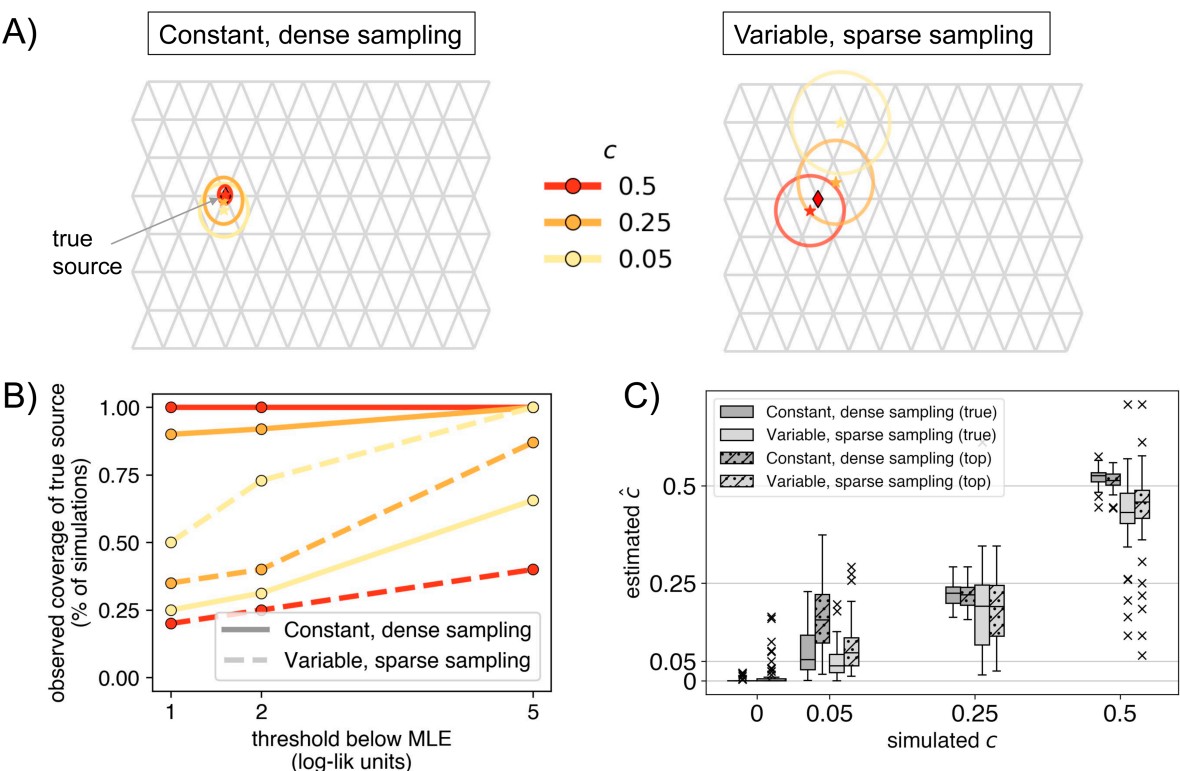

**Fig 2. Range of performance metrics over 50 simulation replicates for each of the two sampling scenarios: A) Mean inferred location represented by a star with paired two-times the standard errors on the mean location represented by boundaries of ellipsoids.** The true source is indicated as a diamond on each grid. **B)** Coverage behavior of the log-likelihood surface, i.e., the percentage of simulations in which the true source was within a certain $x$ threshold of the MLE source location, and **C)** Standard boxplots of the estimated MLE $\hat{c}$ across all replicates for each simulated value of $c$. The plain boxplots show estimated values when the destination is fixed to the true simulated deme and the dashed and dotted boxplot represent estimates when the destination is set to the top outlier.

that this coverage is 100%, $C_{95} = [93\%, 100\%]$ for 2 log-likelihood units, and in the sparse scenario, it is 70%, $C_{95} = [56\%, 83\%]$. The performance of the estimated source fractions is shown in Fig 2C. We estimate a small mean bias in both scenarios (0.031 $[0.018, 0.045]$ and $-0.033$ $[-0.05, 0.015]$, for $c > 0$) with a higher absolute bias in the sparse scenario.

## Applications

### Gray wolves (*Canis lupus*) from North America

We applied the method to the North American gray wolves data set of 111 individuals genotyped across 17,729 SNPs from [19] used in the original FEEMS publication [10] (see Sect S1 in S2 Text for details). North American gray wolves are a highly mobile species [19] that show patterns of population structure consistent with isolation-by-distance [32]. Moreover, from a practical standpoint, this data set also provides a useful testing scenario for our method given the sparsity of the sampling (that is typical for ecological studies conducted in the wild) and the prominent geographical features that introduce spatial heterogeneity in baseline effective migration rates across a broad continental scale. Further, dispersal patterns of these wolves are difficult to study due to the practical complexity in sampling such large areas while also accounting for external factors like seasonal, non-reproductive migrational patterns [33,34].

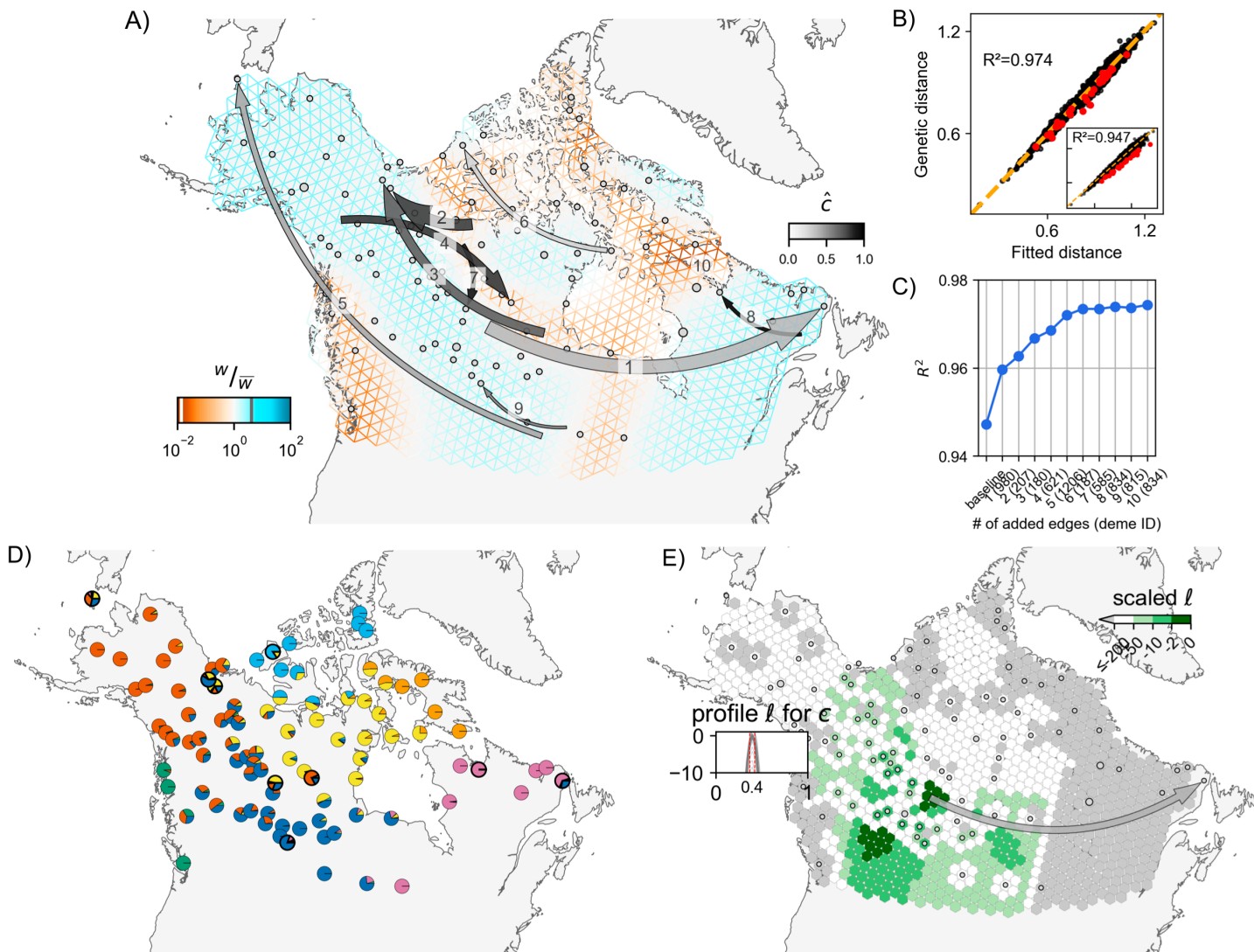

**Fig 3. A, B, C) Full suite of `FEEMSmix` results with** $K = 10$ **for 111 North American gray wolves from** [19]. **D)** The average of individual admixture proportions for each deme from an `ADMIXTURE` $K = 7$ run. The nine demes from subfigure **A** are outlined in black. **E)** Inferred surface for a particular long-range event to destination deme *980* (an example of an outlier that could be explained by admixture proportions) with $\hat{c} \approx 0.4\ [0.37, 0.42]$ from `FEEMSmix`. The parameter $\hat{c}$ reflects the estimated fraction of lineages from a particular source necessary to explain the observed long-range genetic similarity between deme *980* and the MLE source, and the confidence interval is calculated using an asymptotic approximation. The triangular grid is constructed using the `dggrid` package with `res=6` [35], and the base map is drawn using shape files generated by `Cartopy` (with the base layer available at https://www.naturalearthdata.com/download/50m/physical/ne_50m_land.zip, [36]).

The baseline `FEEMS` fit with the deme-specific variances and $(\lambda_{\mathrm{CV}}, \lambda_{q,\mathrm{CV}}) = (2, 10)$ is shown in Fig 3A and S5 Fig (the result is similar to the result in [10] which uses a single fixed variance). The model fits the regions encompassing geographical features like the the coastal mountain ranges in British Columbia, major waterways, and the tundra/boreal forest transition as having lower effective migration (model fit of $R^2 \approx 0.95$).

We ran the *iterative* fitting scheme over $K = 10$ long-range edges, and found 9 unique recipient demes as a result. Each of these unique demes contained just a single individual. The identity of at least three of these recipient demes (*402, 621, 1206*, see S23 Fig for location of these demes) can be explained, as the original [19] study had classified individuals belonging

to these demes as being putatively admixed based on their ancestries from multiple sources in an `ADMIXTURE` analysis [37]. In a similar vein, Fig 3D (also see S6 Fig), shows that 8 of 9 demes identified by `FEEMSmix` (outlined in black) are modeled by `ADMIXTURE` with substantial proportions of ancestry from multiple sources, and each of these 8 demes appear distinctive from their surrounding demes in their ancestry profile. With close inspection, we find the LREs depicted by `FEEMSmix` generally help show pairs of distant populations in the `ADMIXTURE` plot that share ancestry in the same source populations.

Interestingly, LRE 10 shares the same destination as LRE 8, which is the one deme that is not found to have ancestries from multiple sources in an `ADMIXTURE` analysis. Additionally, of our top ten edges, LRE 10 is the only edge that falls below the $L_r \leq 10$ threshold level of support (as indicated by the dashed arrow in Fig 3A).

To understand the inferred LREs in more detail, we examined sample meta-data. We assessed whether batch effects due to shared season of sampling (or year of sampling) could explain any of the long range similarities, but we found no such systematic associations. However, we did find that 5 of 9 outlier samples have clear sampling issues or at least issues that raise concern; i.e., 2 of 9 were found to have recording or clerical errors in their locations, and 3 of 9 were recorded at vague or questionable locations (see Fig 4, and Sect S2.3 in the S2 Text for further discussion).

For comparison, we also ran two previously published methods to better contextualize our results: `SpaceMix` (spatial and with similar methodology for modeling admixture, [18]) and `TreeMix` (non-spatial but with very similar methodology for fitting residuals, [16]) (see Sect S2 in S2 Text for more details). We observed largely overlapping results amongst the three methods but found that `FEEMSmix` produced the largest set of unique long-range "dispersal" events (even when only considering events with strong support, i.e., solid arrows), whereas the other methods only found a subset of these putative events.

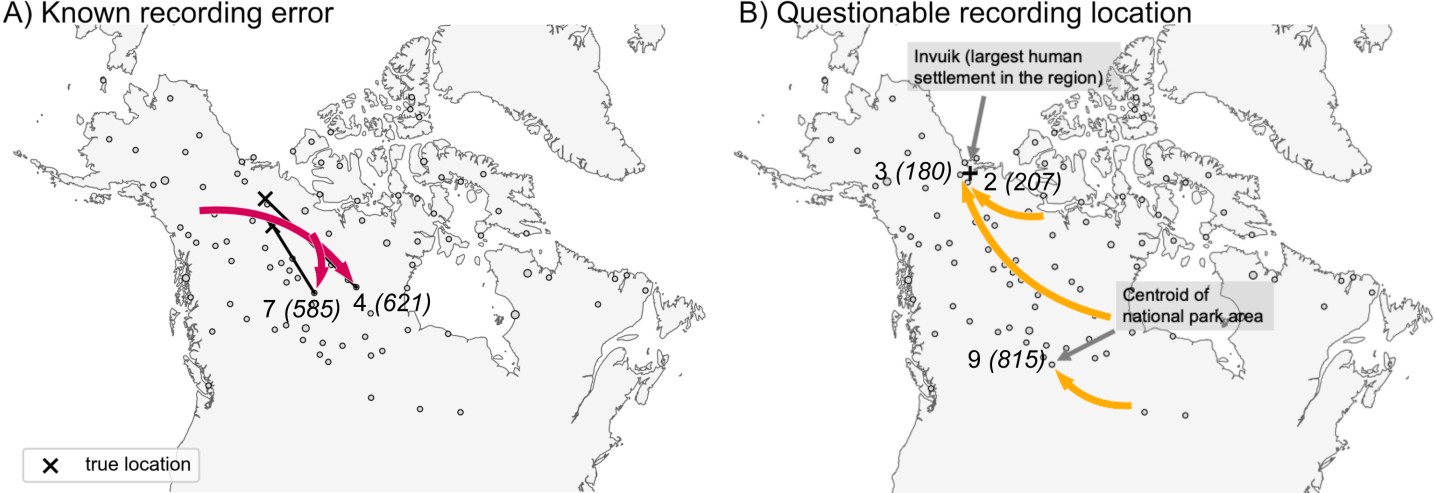

**Fig 4. Results from examining the LREs found by `FEEMSmix` for potential explanatory factors.** The figure shows several LREs labeled by edge ID and deme ID from Fig 3 and interpretations based on an investigation of the sample meta-data (see Sect S2.3 in S2 Text). **A)** For two samples, recording errors were discovered upon close inspection, and 'X' marks the corrected location of each sample. **B)** For three samples, the recorded locations are plausibly central locations where a sample was recorded as originating from, though the actual sampling event took place further away. The base map is drawn using shape files generated by `Cartopy` (with the base layer available at https://www.naturalearthdata.com/download/50m/physical/ne_50m_land.zip, [36]).

With regards to runtime on the wolf data, FEEMSmix took 2.5 hours for the entire workflow (with the baseline fit in FEEMS taking 15 minutes and fitting of the 10 LREs taking a total of 2.3 hours). In comparison, TreeMix took 32 hours to fit 15 edges and SpaceMix took 5 hours when estimating just admixture source locations and 6 hours when estimating both admixture source and geo-genetic locations (recommended usage). Both TreeMix and FEEMSmix will scale proportionally with the number of edges added to the baseline fit whereas the convergence in SpaceMix will depend on the complexity of the sampling posterior. We also note FEEMSmix can be run in parallel if each long-range edge is fit independent of other ones (i.e., in an alternative mode).

**Spatial assignment via leave-one-out cross-validation.** To evaluate the spatial assignment capabilities of our method, we performed leave-one-out cross-validation experiments. For each individual in the dataset, we masked their true geographic coordinates with fixed placeholder locations at the center of the habitat, and then attempted to predict their origin using the FEEMSmix likelihood. Specifically, we fit the masked individual's location as a long-range gene flow event (like in Fig 1, Step 4), with the MLE of the source location serving as our predicted origin. This leverages the method's ability to identify optimal source-destination pairs with applications for spatial assignment of samples of unknown provenance (e.g., museum specimens or retrieved biological material from smuggling, in the style of [38]).

We repeated this procedure for every individual and evaluated assignment accuracy using great circle distance between predicted and true locations. Our results show that FEEMSmix performs comparably to or slightly better than a recently developed deep learning method (Locator, [39]) for the same task (see S7 Fig).

## Humans (*Homo sapiens*) from across Afro-Eurasia

We also applied our framework to 4,070 modern humans from across 319 unique locations in Afro-Eurasia (data available from [40], see Sect S1 in S1 Text for details). This dataset was first compiled by [20], and serves as a useful test bed for our method due to its broad geographic sampling, and more importantly, the ability to corroborate signatures of long-range genetic similarity with prior knowledge from both genetic and non-genetic sources, including archaeology, anthropology, and linguistics [41]

First, we applied FEEMS to this data and found many previously-known patterns in human genetic variation, like elevated differentiation across the Saharan Desert, the Mediterranean Sea, and the Himalayan mountain range (Fig 5A for baseline FEEMS model with $(\lambda_{CV}, \lambda_{q,CV}) = (3, 1)$).

We ran the same *iterative* fitting procedure as before for $K = 10$ LREs and found ten unique recipient demes. All ten recipient demes contain individuals from a single population label in the sample meta-data. Additionally, all ten edges show strong support under our framework with $L_r > 100$.

The resulting ten long range edges fit by the method (shown in Fig 5C) can be interpreted in terms of five major signals, which we discuss in terms of the first added edge for each signal:

1) Aleut individuals from the Kamchatka peninsula in Russia (from [42]) with a source fit from western Russia, with $\hat{c} \approx 0.5$ [0.49, 0.52] (LRE 1). This long-range genetic similarity is supported by previous mtDNA results [43] and ADMIXTURE results which show that about 20% of ancestry can be attributed to an ancestry found in northern Europe in models with $K = 3 - 12$ (from the original publication and S8 Fig). We see similar signals for the Tlingit

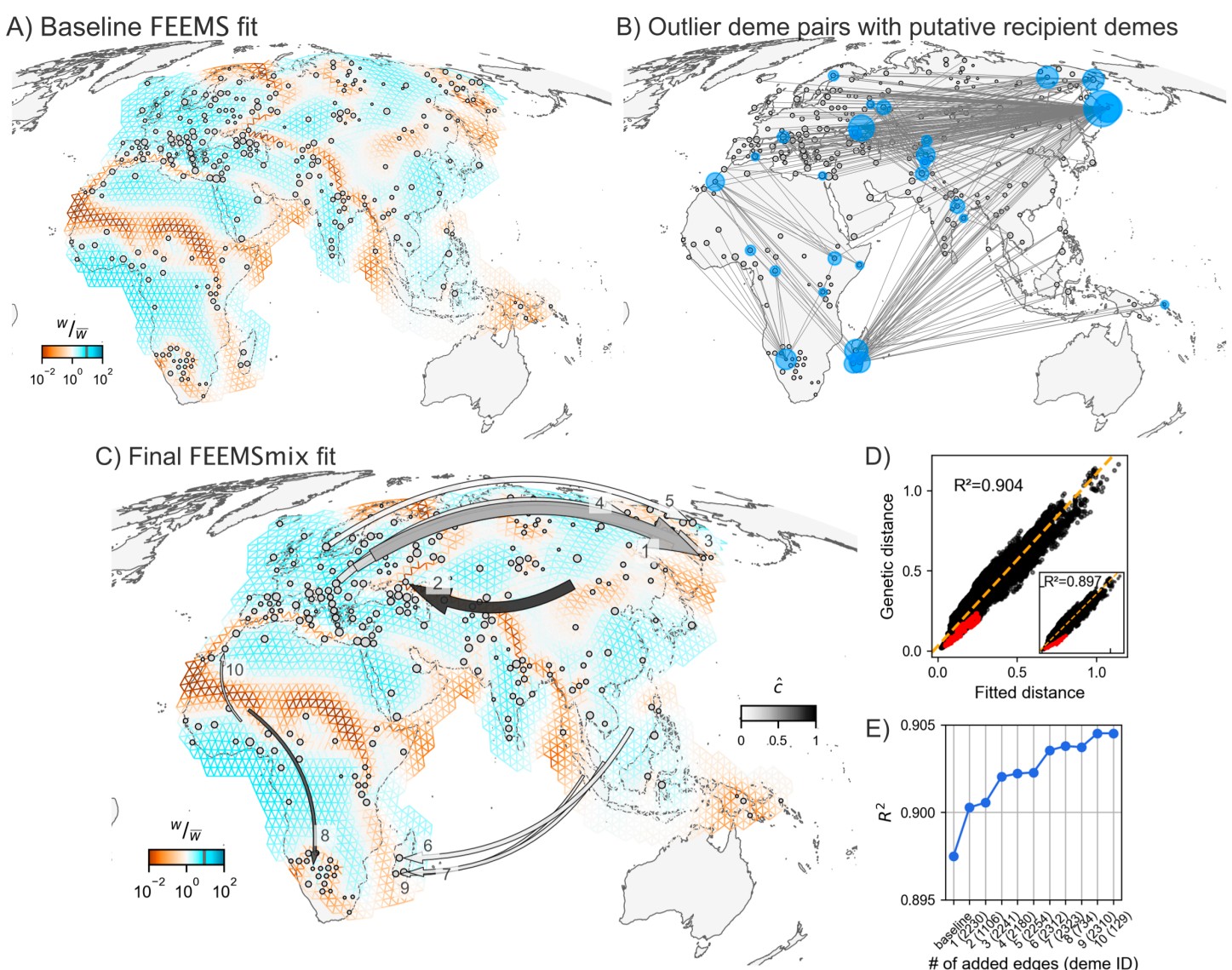

**Fig 5. Empirical results from `FEEMSmix` for 4,070 humans from [20]. A)** Baseline `FEEMS` fit for the 271 sampled demes with inferred migration troughs reflecting areas of historically low migration (e.g., Saharan Desert, Himalayan mountain range, Mediterranean Sea). **B)** A map showing the top 1% of outlier pairs, with putative recipient demes highlighted in blue by `FEEMSmix`. **C, D, E)** Full suite of `FEEMSmix` results with $K = 10$ LREs. The triangular grid is constructed using the `dggrid` package with `res = 5` [35], and the base map is drawn using shape files generated by `Cartopy` (with the base layer available at https://www.naturalearthdata.com/download/50m/physical/ne_50m_land.zip, [36]).

individuals from [42] (LRE 3), Yukagir individuals from [44,45] (LRE 4), and Chukchi individuals from [44,46] (LRE 5) with decreasing strengths of northern European admixture (see [42,47], and S8 Fig);

2) Kalmyk individuals from eastern Russia (from [45]) with a source fit from the region of Mongolia and $\hat{c} \approx 0.8\ [0.77, 0.82]$ (LRE 2). Kalmyks are a Mongolic-speaking group residing in Europe, with recent origins from East Asia with the latest wave of migration reported as happening in the 17th century [48];

3) Vezo individuals (north-western deme) from Madagascar (from [49]) with source fit in southeast Asia with $\hat{c} \approx 0.12\ [0.1, 0.13]$ (LRE 6). This LRE aligns with a well-known ancestral

long-range dispersal across the Indian Ocean from south-east Asia with plenty of supporting genetic [49,50] and linguistic [51,52] evidence. We see similar geographic signals with similar strengths replicated for the other Madagascar populations as well (Antemoro in the south-eastern deme (LRE 7) and Mikea in the south-western deme (LRE 9));

4) Bantu Herero individuals from Botswana or Namibia (from [42]) show a source from western Africa with $\hat{c} \approx 0.73$ $[0.71, 0.74]$ (LRE 4). This signal is also backed by an `ADMIXTURE` analysis at $K = 13$ that shows a western Bantu component maximized in these individuals in [53];

5) Moroccan individuals from south Morocco (from [54]) a with a source fit proximal to coastal Ghana and Nigeria with $\hat{c} \approx 0.23$ $[0.21, 0.25]$ (LRE 10). [54] found that the southern Moroccan individuals are closest to the Luhyan population from Kenya (this relationship is also replicated in the initial outlier pairs found by `FEEMSmix` for this deme in Fig 5B). However, [54] also infers that a migration to Morocco from sub-Saharan Africa occurred about 1,200 years ago, which coincides with the rise of the Ghanaian Empire that was involved in the trans-Saharan slave trade. This claim has gained further support more recently using haplotypic segments in [55] — and perhaps corroborates the placement of a source in western Nigeria from `FEEMSmix`.

Finally, the outlier deme pairs and the inferred source locations from `FEEMSmix` (in Fig 5B and 5C) are also supported by the geographic distribution of admixture $f_3$ statistics (shown in S9 Fig for a few selected demes, computed using `AdmixTools` from [56]).

## Discussion

In this paper, we present a method called `FEEMSmix` that represents the geographic structure of genetic variation using simultaneously a landscape of spatially heterogeneous gene flow and long-range gene flow events. It is built upon a previous method called `FEEMS` (Fast Estimation of Effective Migration Surfaces by [10]), and follows in the same naming tradition of modeling residuals to baseline fits of the observed genetic data with a parameter specifying the strength of an instantaneous admixture pulse (e.g., `TreeMix`, [16]; `MixMapper`, [17]; `SpaceMix`, [18]). However, we note that since we use the same approximate coalescent-time-based likelihood formulation as in `EEMS` [9], this method could also rightly be called `EEMSmix`.

We have examined the sensitivity and accuracy of this method in simulations that mimic real-world settings, most importantly extremely sparse sampling. We also demonstrated the ability of the method to recover known signals of long-range gene flow events in a large empirical data set of humans across an Afro-Eurasian panel. While we focused on $K = 10$, we do not think that these are the only instances of long-range dispersal in human history. For example, we find many instances of previously understood long-range genetic similarities for values of $K$ between 10 and 25 (all of which show significant support for long-range gene flow, see S10 Fig), e.g., Tiwari Brahmin from UP, India (LRE 12 shows a high proportion of Iranian ancestry, [57]), Hazara from Afghanistan (LRE 20 shows Central Asian ancestry, [42]), and Masai from Kenya (LRE 21 shows Nilotic ancestry, [42]).

In our analysis of 111 wolf samples across North America, we detect new signals of long-range genetic similarity, some of which appear to be genuine long-range dispersal, and others which appear to be artifacts of sampling. In many data sets one might expect small observational or recording errors to be on a scale that is negligible compared to patterns of isolation-by-distance and inconsequential for most analyses. Here, the method helped identify two cases of mislabeled samples because they were identified as having unexpected long-range

genetic similarities. In another case within the wolf dataset, one destination deme (with a single individual, sampled at Latitude = 58.16; Longitude = –68.42) is implicated by two edges, LREs 8 and 10. For LRE 8, the source fraction estimated is high $\hat{c} \approx 0.9\ [0.88, 0.91]$ from a source located within a known migration corridor, indicating high genetic similarity to neighboring individuals (also evidenced by its placement on a tree inferred by `TreeMix` in S11 Fig). LRE 10 shows little support ($L_r \leq 10$). Furthermore, the deme implicated by LRE 8 and 10 shows ancestry from a single source in a complementary `ADMIXTURE` analysis. Based on these multiple lines of evidence, we conclude that there is little support for long-range gene flow to this individual, suggesting that this represents a "false positive". We advice users to follow a similar pattern of reasoning when interpreting the results from our method.

As hinted, we find the method complements existing methods like `ADMIXTURE`, `TreeMix` and `SpaceMix` with a chief advantage being that `FEEMSmix` is unique in providing a single integrated framework for explicitly using geographic information to model varying rates of local migration as well as long-range admixture events. Running these other methods in a complementary fashion with `FEEMSmix` can help with corroborating results and gaining a more robust understanding of the genetic structure in one's data.

As with any method that summarizes complex data using simple models and the notion of effective parameters, our method also comes with limitations that influence the interpretation of our results. First, our method `FEEMSmix` is built on top of `FEEMS` [10], so it shares the same underlying assumptions, which, in turn, influence the outliers that are chosen for analysis in `FEEMSmix`. Briefly, `FEEMS` requires a choice for the overlaid nearest-neighbor graph ("grid") of assumed demes. It fits stationary, symmetric migration rates to the data, which can have drawbacks when asymmetric gene flow is pervasive (see [15,58]).

Operationally, in terms of data, the method assumes the input are genotypes with no missing data and that all sampling locations are resolvable to the scale of the input grid of assumed demes. `FEEMS` also assumes the genetic data have some form of geographic structure to begin with, and in the supplement we show examples of how `FEEMS` and `FEEMSmix` perform on data from a panmictic population (S12 Fig); and remind users to first assess whether geographic structure exists before applying `FEEMSmix` (e.g., by examining the relationship between pairwise geographic and genetic distances directly, see S12B Fig).

The choice of tuning parameters $(\lambda, \lambda_q)$ will affect the results from `FEEMSmix` in two ways: 1) the identity of potential LREs being implicated in the baseline `FEEMS` fit, and 2) the estimate of the resulting source fraction. As best practice, we recommend trying out different values of $\lambda$ and $\lambda_q$ spaced evenly on a log-scale around the baseline $\lambda_{CV}$ value (e.g., $\{0.05 \times \lambda_{CV}, \lambda_{CV}, 20 \times \lambda_{CV}\}$), when fitting the data. Typically, most outlier demes will persist through these different settings, but observing how the results change can be informative about the underlying signals in the data (see S3–S4 Figs for replicates in simulations, and S5 Fig and S13 Fig for results with the empirical data sets). In simulations and in empirical data, the effect of changing $\lambda_q$ is often negligible on both the outlier detection and the resulting source fraction estimates relative to the effect of the tuning parameter on the migration weights (see S14 Fig for a comparison of this in simulations).

Defining rigorously statistically justifiable stopping criteria for adding LREs, akin to the challenge of choosing $K$ in admixture models [59], is particularly challenging in this problem setting given the use of penalized likelihoods. While the parametric bootstrap represents a statistically principled approach, the computational overhead involved in the iterative process makes it impractical for the size of typical datasets used in `FEEMS`. We explored how the change in $R^2$ and $L_r$ behave as indicators the support for adding LREs, and we find both are helpful, especially when the signal for an LRE is strong; yet when subtle model violations exist

or there is extreme uneven sampling (e.g., S19 Fig, S21 Fig and S22 Fig), we find these heuristics can be misleading. We therefore recommend that users use $L_r$ as a guide rather than a definitive test, particularly in scenarios involving multiple overlapping migration events. As is standard with `TreeMix`, we encourage users to examine multiple edges and to articulate in their interpretation the rationale for the choice and possible limitations in the final displayed value of $K$ used in any analysis.

In cases where the true history is not of an instantaneous gene flow event, the estimated source fraction $\hat{c}$ should be viewed as an *effective* parameter that simply reflects the source fraction necessary to model the residual genetic similarity while accounting for a background of spatially heterogeneous local dispersal. In the case of an older pulse of gene flow, the expectation is, and our simulations show, an attenuation of signal over successive generations of background gene flow, causing $\hat{c}$ to be downward biased (see S15–S16 Figs). Also, if there are multiple events from disparate sources to the same destination, `FEEMSmix` will typically identify first the source with the most concentrated geographic signal, though the log-likelihood surface could weakly reflect the presence of multiple sources depending on the background rates of gene flow (see S17 Fig). In cases, where a particular source contributes lineages to multiple neighboring destination demes (as implied by the fitted human data, see LREs 1, 3, 4, 5 and LREs 6, 7, 9 in Fig 5C), we find that `FEEMSmix` successfully detects *each* event separately in ∼80% of simulation replicates (see S21–S22 Figs).

The effects of geographic sampling biases are important to consider. Conceptually, sampling a geographic range need not be perfectly exhaustive to infer major gene flow features across a geographic range, because high or low regions of gene flow are likely to impact on the movement and coalescent rates of the ancestral lineages of sampled individuals nearby. However, features that are not "experienced" by the ancestors of a sample will go undetected, and this is an important caveat. For `FEEMSmix`, if a long-range gene flow event is truly recent and between unsampled demes, one will not be able to detect the event in the data; however, if there has been sufficient time since the long-range gene flow event for the source lineages to traverse between unsampled and sampled demes, then `FEEMSmix` may potentially detect the event indirectly, depending on the background migration rates and time since event. As an example, in simulations with geographically biased sampling schemes, including a scenario where an entire portion of the habitat is left completely unsampled, we find adding a small number of samples in a previously unsampled region is sufficient for `FEEMSmix` to accurately identify the strength and direction of specific long-range gene flow events (see S18–S20 Figs) even as the underlying migration surface remains uninformative under such sparse sampling. Overall, with these considerations in mind, we generally recommend multiple sampling locations chosen uniformly across the habitat over a smaller number of well sampled locations.

A compelling extension of the method would be add more precise modeling of older gene-flow events, though this would require modeling the migration of lineages across the landscape, which adds computational complexity and may be more sensitive to violations from equilibrium assumptions. Another interesting extension would be to model gene flow as arising from a region rather than a single point source as is done here. One could imagine a model in which a set of sources contribute some lineages to a certain destination deme or set of demes. However, fitting such models would require searching over a large space of possible parameters, and it raises the issue of power in the data to discern among multiple plausible scenarios. One could also envision modifying the existing `FEEMS` framework by removing short-range edges and adding LREs in a more expansive form of migration graph inference. For example, in regions where gene flow occurs primarily through occasional long-range dispersal events across an unsuitable habitat, a small number of LREs could replace multiple

short-range edges, providing a more biologically realistic model. This approach would be particularly relevant for scenarios like trans-oceanic human migrations or dispersal across large geographic barriers where intermediate stepping-stone populations are absent.

Overall, the ideal framework for understanding the genetic structure of any species would infer time-varying dispersal regardless of its spatial scale with just a sparse sampling of individuals across the range. Though the method presented here does not achieve this ideal goal, it takes a step in this direction, and provides a useful tool for describing the geographic structure of genetic variation by simultaneously illuminating long-range genetic similarity over a background of spatially heterogeneous patterns of isolation-by-distance.

## Supporting information

**S1 Text. Features of empirical data.**
(PDF)

**S2 Text. Results from North American gray wolf samples of Schweizer et al. (2016).**
(PDF)

**S1 Fig. A visualization of the two sampling schemes from the main text for a single simulation replicate.** The gray circles represent sampled demes on the overlaid triangular grid (as is custom in FEEMS).
(TIF)

**S2 Fig. The *constant, dense sampling* analog to Fig 1 in the main text.** With dense sampling, we see that the baseline FEEMS fit works well to capture the central barrier in the grid, with FEEMSmix also accurately estimating the source and strength of the long-range event. But we also see that adding too many edges can lead to overfitting with a decrease in model $R^2$ with $K = 3$ LREs.
(TIF)

**S3 Fig. FEEMSmix results for 15 simulation replicates from the *constant, dense sampling* scenario.** In **A)**, we observe that the true destination deme is always implicated as the first LRE in each replicate. This is true over all 50 simulation replicates. In **B)**, we see a systematic increase in model $R^2$ with the first (true) LRE and a subsequent plateau with more added LREs. In general, over 50 simulation replicates, we find that $L_r > 10$ in 95%, $C_{95} = [86\%, 100\%]$ of replicates for $K = 5$ edges indicating that a mixture model of two Normals best fits the residuals with this sampling strategy (see S2 Fig).
(TIF)

**S4 Fig. FEEMSmix results for 15 simulation replicates from the *variable, sparse sampling* scenario.** In **A)**, we observe that the true destination deme is implicated as the first LRE in each replicate, while the other LREs are random with respect to the geographical source. In **B)**, we see a systematic increase in model $R^2$ with the first (true) LRE and a subsequent plateau with more added LREs. In general, over 50 simulation replicates, we find that the first LRE always captures the direction and strength of the simulated event even with such sparse sampling. We find that in 94%, $C_{95} = [83\%, 99\%]$ of all simulation replicates the first edge has $L_r > 10$, while only 30%, $C_{95} = [17\%, 44\%]$ of the simulation replicates show $L_r > 10$ for the second LRE, dropping down to 5%, $[0\%, 14\%]$ for the third LRE.
(TIF)

**S5 Fig. Baseline fits and residual outlier pairs from the three modes for how the variance parameters are modeled in FEEMS:** *None* refers to a single variance parameter that is

fixed at a value estimated by a model assuming a single weight across the entire grid [default in10, comes from an initialization step before optimizing the weights, ∼2.1s], '*1-dim*' refers to a single, estimated variance parameter that is jointly estimated with all the other weights in the graph (∼13.5s), and '*n-dim*' refers to estimating a variance parameter for each sampled deme (default in FEEMSmix, ∼11.3s). We observe that we obtain better fits with increasing number of parameters in the framework (based on $R^2$) though with increasing runtime. However, visually, all three methods pick up the major barriers and corridors in the data set. But, an important point to note here is how the pinwheel-like patterns around sampled demes disappear with the estimation of deme-specific variance parameters. Also, the outlier demes implicated in the fits change on a gradient between these modes: with Arctic and HighArctic demes being 75% of outliers in *None* to just 20% in '*n-dim*'. The base map is drawn using shape files generated by Cartopy (with the base layer available at https://www. naturalearthdata.com/download/50m/physical/ne_50m_land.zip, [36]).
(TIF)

**S6 Fig. Average admixture fractions observed within each sampled deme from an ADMIXTURE [37] analysis from** $K = 2 - 8$**.** The base map is drawn using shape files generated by Cartopy (with the base layer available at https://www.naturalearthdata.com/download/50m/ physical/ne_50m_land.zip, [36]).
(TIF)

**S7 Fig. Comparison between FEEMSmix and Locator (a deep-learning based method from [39]) in predicting spatial locations of samples from the wolves data set in a leave-one-out based approach.** We see comparable results between the two methods, with slightly better performance in FEEMSmix (decrease in median error of approx. 100 km, though high error with both methods indicate how mobile wolves tend to be). True sample locations are shown as black points and the predicted locations are shown in the color corresponding to each method. An interesting point to note here is that the predicted locations from either method never cross the migration barriers as estimated by FEEMS, also indicating a correlation in predicted location (average cosine similarity of ∼0.5). The base map is drawn using shape files generated by Cartopy (with the base layer available at https://www.naturalearthdata.com/ download/50m/physical/ne_50m_land.zip, [36]).
(TIF)

**S8 Fig. The top 25 LREs from FEEMSmix overlaid on a map with ADMIXTURE results from a** $K = 7$ **analysis.** The base map is drawn using shape files generated by Cartopy (with the base layer available at https://www.naturalearthdata.com/download/50m/physical/ne_50m_land. zip, [36]).
(TIF)

**S9 Fig. Admixture** $f_3(A, X; T)$ **statistics for selected putative destination demes identified by FEEMSmix (values computed using v651 of AdmixTools from [56]).** Here, $A$ is the reference population, $X$ is a testing population and $T$ is the target population (i.e., destination deme). We observe that populations with the most negative $f_3$ values (dark red) in each case are found in similar locations to the paired source demes inferred by FEEMSmix (in Fig 5B and 5C in the main text). The location of $T$ is displayed as a green diamond and the location of the corresponding $A$ is shown as a pink square. The base map is drawn using shape files generated by Cartopy (with the base layer available at https://www.naturalearthdata.com/ download/50m/physical/ne_50m_land.zip, [36]).
(TIF)

**S10 Fig. Full suite of FEEMSmix results from the human data set with** $K = 25$ **LREs.** The base map is drawn using shape files generated by Cartopy (with the base layer available at https://www.naturalearthdata.com/download/50m/physical/ne_50m_land.zip, [36]). (TIF)

**S11 Fig. TreeMix [16] results with** $m = 15$. (TIF)

**S12 Fig. FEEMSmix results for 15 simulation replicates from the** *variable, sparse sampling* **scenario under panmictic conditions.** In **A)**, we observe that FEEMS estimates a flat migration surface across all replicates, as expected. Out of 30 simulation replicates, the top LRE is found to show $L_r > 10$ in 35%, $C_{95} = [22\%, 50\%]$ of all simulations, and we note that fitting the model to data with panmixia rather than isolation-by-distance makes evaluation of this outcome difficult (in some sense migration is all long-range in a panmictic population). In **B)**, we show that $\hat{\beta} \approx 0$ $[-0.0006, 0.0006]$ across all simulation replicates, reinforcing that there is no geographic structure in the genetic data. (TIF)

**S13 Fig. A range of fits to the human data set across different tuning parameter values with the top** 1% **of outliers displayed using the method presented in the main text for human data.** With decreasing $\lambda$ values, we see new demes being implicated as outliers in the fit, that disappear with a very low $\lambda$ value. This is likely due to overfitting with low $\lambda$, i.e., the baseline FEEMS fit connects two distant but similar demes via a snaking migration corridor connecting them, hence modeling away any residual. For instance, the Kalmyk individuals are no longer found as outliers, as they are connected to individuals further east through a migration corridor that punctuates a region of low inferred migration that is inferred with higher $\lambda$ values. As expected, the baseline model $R^2$ increases with decreasing $\lambda$, which is why leave-one-out cross validation is used to choose the optimal value for the $\lambda$'s. The base map is drawn using shape files generated by Cartopy (with the base layer available at https://www.naturalearthdata.com/download/50m/physical/ne_50m_land.zip, [36]). (TIF)

**S14 Fig. The effect of various tuning parameter values** $(\lambda, \lambda_q)$ **on the estimated admixture proportions** $\hat{c}$ **under the two sampling scenarios.** The true simulated strength is marked as a red line at 0.25. We observe that there is little to no difference in estimated values across different tuning parameter combinations in the case of *constant, dense sampling*, but quite a big difference in the case of *variable, sparse sampling*. Essentially, for very small values of $\lambda$, FEEMS overfits the patterns in the data (see the more disjoint maps in S4 Fig for a visualization of this), leading to very little extra residual, and in turn, leading to an underestimation of the strength of long-range gene flow. The most common pair of tuning parameter values chosen by the cross-validation procedure was (20.0,100.0) for the dense scenario (similar to (20.0,20.0) in the figure) and (0.3,1.0) for the sparse scenario (similar to (1.0,1.0) in the figure). (TIF)

**S15 Fig. Fitting an older admixture event: Here we show the results of using FEEMSmix on data simulated under an older admixture pulse 10 generations ago with** $c = 0.5$. Older admixture events are expected to diffuse the long-range migrant lineages across the landscape and leave a much less concentrated spatial signal. In **A)**, we see the mean inferred location (a ★) and two-times the standard errors of the mean (an ellipse) for the two sampling scenarios presented in the main text (compare to Fig 2A). The true source in the simulations is

represented by the red cross. In **B)**, we show the coverage statistics and see that the coverage statistics are lower in the case of an older admixture event (compare to Fig 2C). In **C)**, we see the model estimates a smaller admixture proportion than the one simulated, likely because the signal is expected to decay over time due to the ongoing background migration on the landscape.
(TIF)

**S16 Fig. Fitting older admixture events: across 30 simulation replicates of older admixture pulses ($\tau = [2, 10, 20, 100]$ generations ago) of $c$ = 0.5 over a uniform surface of weak $\overline{Nm}$ = 0.01 and strong $\overline{Nm}$ = 0.1 background migration.** Overall, with increasing background migration and older admixture pulse, there is a washing out of long-range gene flow signal (as expected, see also S15 Fig). In **A)**, we see that the model's ability to pick the true destination deme as the top outlier drops off after a certain time back in the past. In **B)**, we see that in certain regimes, the true deme is not implicated in the top 10 largest outliers as found by `FEEMSmix`, indicating a loss of the signal. In **C)**, we see that with weak background migration, the model can pick out the true source of the long-range gene flow in about ∼80% of the replicates when the true destination is implicated. However, this performance drops off sharply under a higher level of background migration. Finally, in **D)**, we see the characteristic decay in the estimates for the source fraction for older admixture pulses and stronger background gene-flow due to the diffusion of lineages across the surface (similar to S15 Fig).
(TIF)

**S17 Fig. Result from a simulation with two geographically distant *unsampled* sources to the same destination over a uniform migration surface.** In **Ground truth**, we show that there are two simulated long-range gene flow events with differing strengths (NW source: $c_1$ = 0.5, SE source: $c_2$ = 0.25) from diagonally opposite parts of the habitat. We see the two regions with the true sources (in red crosses) is found exactly by `FEEMSmix` across the first two LREs with low error ($\hat{c}_1 \approx 0.45, \hat{c}_2 \approx 0.3$) on adding $K$ = 5 edges. Though, we also observe that in this setting, the $L_r$ statistic does not show large values as one might for two true LREs.
(TIF)

**S18 Fig. Simulation replicates showing $K$ = 3 LREs in the case when there is inhomogeneous sampling of the habitat, more specifically, there is biased sampling in a manner in which the barrier is not sampled.** Despite the sampling bias, `FEEMS` does pretty well in capturing this area of low migration. `FEEMSmix` has both the source and the destination demes sampled and we see that in each replicate, the correct source, destination, and strength of direction of long-range gene flow is inferred. This is true across all 50 simulation replicates, and only ∼25%, $C_{95} = [13\%, 40\%]$ of all replicates show a second LRE with $L_r > 10$.
(TIF)

**S19 Fig. `FEEMSmix` results with $K$ = 3 LREs for 15 simulation replicates with a strongly biased sampling strategy, wherein only the east side of the habitat is sampled.** This presents an extremely challenging case. In **A)**, we observe that the simulated long-range event is *never* captured, though the existence of a 'pinwheel'-like pattern around the destination deme indicates an interesting signal. Additionally, `FEEMS` also fails to capture the migration corridor on the west side of the habitat. In **B)**, we observe that adding any number of edges barely increases the fit to the data. Without sampling the source, there is apparently little gain to add an LRE in terms of $R^2$. All added LREs pass the $L_r > 10$ threshold as there is still systematic structure that is not captured on the habitat.
(TIF)

**S20 Fig.** `FEEMSmix` **results with** $K$ = 3 **for 15 simulation replicates with a softly biased sampling strategy, wherein the east side of the habitat is sampled densely with a sparse random sampling (only ∼5%) on the west side.** This also presents a very challenging case, but slightly less than in S19A Fig. In **A)**, we observe that the simulated long-range event is *always* captured as the top event, indicating that in this setting even a small amount of sampling is sufficient to detect the long-range event. However, `FEEMS` still fails to capture the migration corridor on the west side of the habitat. In **B)**, we observe that adding the first LRE provides a significant improvement in fitting the data with a plateau observed after $K$ = 1 (similar to S4B Fig and improved compared to S19B Fig). With regard to the LREs, here too, we observe a similar behavior as in S19 Fig wherein all edges show $L_r > 10$.
(TIF)

**S21 Fig. Simulation replicates showing** $K$ = 5 **LREs in the case when three adjacent demes share the same admixture source and lie within an area of** *low* **effective migration.** This is meant to mimic the scenario implied by LREs 1, 3, 4, 5 (eastern/northern Europe to western Russia/Siberia) in the human data. The true source is marked as a red cross on the grid. Across all 50 simulation replicates, we find that there is at least one edge, with over 95%, $C_{95}$ = $[86\%, 100\%]$ of simulation replicates showing at least two edges, reflecting the simulated direction and strength of the three long-range gene flow events. However, in this case, we find that only 30%, $C_{95}$ = $[19\%, 45\%]$ of total replicates show at least one simulated edge with $L_r > 10$, and only 2%, $C_{95}$ = $[0\%, 11\%]$ of replicates showing all three edges with $L_r > 10$.
(TIF)

**S22 Fig. Simulation replicates showing** $K$ = 5 **LREs in the case when three adjacent demes share the same admixture source and lie within an area of** *high* **effective migration.** This is meant to scenario suggested by by LREs 6, 7, 9 (South-east Asia to Madagascar) in the human data. The true source is marked as a red cross on the grid. In this scenario, we find that the three simulated long-range edges are implicated in all 50 simulation replicates across the $K$ = 5 LREs. However, only 25%, $C_{95}$ = $[13\%, 40\%]$ of total replicates show at least one simulated edge with $L_r > 10$.
(TIF)

**S23 Fig. Two plots to aid in interpretation of the** `FEEMSmix` **and** `SpaceMix` **results, and in identifying demes that are picked as recipient demes by these two methods (marked by black arrows).** Deme *576* (also marked with a black arrow) might visually appear to be an outlier here, but its placement in PC space is in accordance with its geographic position.
(TIF)

**S24 Fig.** `SpaceMix` **outputs showing pairwise sample (observed) and parametric (fitted) covariance across the two modes (A: "model 3" and B: "model 4") of running the method mentioned in the text.** Both modes produce $R^2 > 0.9$, and show a somewhat step-like pattern in the fits.
(TIF)

**S25 Fig.** `SpaceMix` **result when estimating both geogenetic and admixture source locations ("model 4").** We see that the two spatial axes of location places these demes in a manner similar to PC1 & PC2 in S23B Fig. The location of the sampled demes correlates with their 'Ecotypic' classification (in S23A Fig). Only deme *737* is implicated in an admixture event, with the location of the source extending over the entire habitat. In *all* cases, we see that the 95% credible intervals for the source locations span the entire 'geo-genetic' space with the

maximum a posteriori estimates differing from the results from running a different mode in S26 Fig.
(TIF)

**S26 Fig.** `SpaceMix` **result when estimating only admixture source locations, keeping sampling coordinates fixed ("model 3").** In the inset plot, we see similar results with `FEEMSmix`, both in terms of the identity of the putative destination demes and the location of the sources of these admixture events. This similarity is also consistent with the results from `ADMIXTURE`. We also display the posterior distributions of the admixture proportions (as boxplots) of the events shown in ellipses in the inset plot.
(TIF)

**S27 Fig. Full suite of** `FEEMSmix` **results from the wolves data set from [19] with** $K = 25$ **LREs.** The base map is drawn using shape files generated by `Cartopy` (with the base layer available at https://www.naturalearthdata.com/download/50m/physical/ne_50m_land.zip, [36]).
(TIF)

**S28 Fig. Full suite of** `FEEMSmix` **results with** $K = 10$ **from a (re)analysis of the wolf samples from [40] on correcting the misreported locations and removing the ambiguous samples.** The base map is drawn using shape files generated by `Cartopy` (with the base layer available at https://www.naturalearthdata.com/download/50m/physical/ne_50m_land.zip, [36]).
(TIF)

## Acknowledgments

We would like to thank members of the Berg, Novembre, and Steinrücken labs, as well as members of the University of Chicago Program in Computational Biology (PCB) community for helpful discussions and feedback during the development of this project. We would also like to thank Rena Schweizer for help with a preliminary interpretation of the results. Additionally, we thank Jeremy Berg, Matthias Steinrücken, and Xuanyao Liu for support at all stages of this work. Computing was performed on servers maintained by the University of Chicago Research Computing Center.

## Author contributions

VS and JN formulated the model. VS ran the simulations and analyses. VS, MM and JN interpreted the results and wrote the manuscript.

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
