## [Decision Letter · Decision Letter 0]

21 Apr 2025

PGENETICS-D-25-00146

Jointly representing long-range genetic similarity and spatially heterogeneous isolation-by-distance

PLOS Genetics

Dear Dr. Novembre,

Thank you for submitting your manuscript to PLOS Genetics. After careful consideration, we feel that it has merit but does not fully meet PLOS Genetics's publication criteria as it currently stands. Therefore, we invite you to submit a revised version of the manuscript that addresses the points raised during the review process.

Please submit your revised manuscript within 60 days Jun 20 2025 11:59PM. If you will need more time than this to complete your revisions, please reply to this message or contact the journal office at plosgenetics@plos.org. Please include the following items when submitting your revised manuscript:

We look forward to receiving your revised manuscript.

Kind regards,

Gideon S. Bradburd

Guest Editor

PLOS Genetics

Kelly Dyer

Section Editor

PLOS Genetics

Aimée Dudley

Editor-in-Chief

PLOS Genetics

Anne Goriely

Editor-in-Chief

PLOS Genetics

**Additional Editor Comments :**

I have obtained 3 expert reviews, which are included below. I agree with the reviewers that the proposed method is a nice extension to existing work and that it would be a useful tool in the empiricist's spatial population genetic toolkit. Although I ask you to respond to each reviewer comment, a few points deserve special attention as you revise. First, I think more could be done to elevate this work and increase its impact by making it of interest to a broader audience. This is a point that came up in several reviews, and I think there are a number of ways to do it, including broadening the scope of the Introduction or elaborating on the method to incorporate a temporal component to the long-distance edges. Another point that all reviewers brought up is that more exploration of the statistical behavior of the addition of LREs is warranted. How should a user know when to stop adding edges? What should we make of the jumps in likelihood that come from adding certain LREs. And, an additional point - what about LRE "false positives?"

I look forward to receiving an updated version of the manuscript!

Sincerely,

Gideon Bradburd

University of Michigan

**Journal Requirements:**

2) Please ensure that your article adheres to the standard Methods article layout and order of Abstract, Author Summary, Introduction, Description of the Method, Verification and Comparison, Applications, Discussion, Acknowledgements, References, and Supplementary Information. For details on what each section should contain, see our Methods article guidelines:

https://journals.plos.org/plosgenetics/s/submission-guidelines#loc-manuscript-organization.

Potential Copyright Issues:

i) Figures (3-5), (S4-S10), S14A, S19A, and S20A. Please (a) provide a direct link to the base layer of the map (i.e., the country or region border shape) and ensure this is also included in the figure legend; and (b) provide a link to the terms of use / license information for the base layer image or shapefile. We cannot publish proprietary or copyrighted maps (e.g. Google Maps, Mapquest) and the terms of use for your map base layer must be compatible with our CC BY 4.0 license.

6) In the online submission form, you indicated that your data will be submitted to the Dryad database upon acceptance. Should your submission be accepted, we will require the following information in your Data Availability Statement:

1. The DOI provided by Dryad

2. The citation for your data package in the reference section of your manuscript

3. The citation for your data package in the methods section

If you are unable to adhere to our open data policy, please kindly revise your statement to explain your reasoning and we will seek the editor's input on an exemption. Please be assured that, once you have provided your new statement, the assessment of your exemption will not hold up the peer review process.

Note : Please check the Data Availability Statement provided in the online submission form and ensure that it includes clear references to the datasets.

7) Please amend your detailed Financial Disclosure statement. This is published with the article. It must therefore be completed in full sentences and contain the exact wording you wish to be published.

8) Please ensure that the funders and grant numbers match between the Financial Disclosure field and the Funding Information tab in your submission form. Note that the funders must be provided in the same order in both places as well. Currently, the order of the grants is different in both places.

**Reviewers' comments:**

Reviewer's Responses to Questions

Reviewer #1: This is an interesting and well-presented manuscript describing an extension to

the EEMS/FEEMS method called FEEMSmix. FEEMSmix models long-range genetic

similarity among samples by identifying outlier residuals from a FEEMS fit and

adding additional edges to the FEEMS topology along which long-range gene flow

can occur. FEEMSmix fills a definite need for describing spatial patterns in

genetic variation data, and I think it will be a welcome addition to the

spatial population genetics toolkit.

I think the big concern with this type approach is identifying which of the long

range edges (LRE) inferred by FEEMSmix represent historical long-range gene

flow events and which represent overfitting by the model. In many cases the

amount of variance explained by a FEEMS model is quite high (90 percent or more

for both wolves and humans), and in those circumstances it seems like there is

a danger of FEEMSmix fitting parameters (=LRE) to random noise in the data. The

authors provide reasonable, corroborated explanations for some of the LRE

identified in their case studies. But presumably there are many LRE for which

there are no obvious explanations, and FEEMSmix has the capability to keep

adding LRE until the model fits the data perfectly. What do we do with all

these LRE? How do we tell which LRE warrant inferences of long-range migration

and which do not? Are there formal model comparison tests that are applicable

and that could be used to compare FEEMSmix to FEEMS to help decide if the

additional complexity added by FEEMSmix is justified on information theoretic

grounds?

I don't think this issue detracts from the general utility of the model, but I

think more discussion and guidance from the authors would be useful for

potential users of the method, perhaps when they are discussing additional

caveats of FEEMSmix in the discussion.

Minor comments:

Line 233, what does "100% [93%, 100%]" mean? This style of reporting appears

throughout this section and I found it confusing. Is it the most frequent value

with the range of values appearing in brackets? Is it an estimated mean and

confidence interval?

Line 241, how well is the true source identified when the destination deme

is *not* fixed to the true one? It seems like a better performance metric would

be setting the destination deme to whatever FEEMSmix thinks is the most

likely.

Eq 1: can you elaborate here? You have written T'[sd] = cT[sd] + (1-c)T[ss] but

my brain wants it to be T'[sd] = (1-c)T[sd] + cT[ss]. What am I missing? Also,

these apply only to sampled demes, is that correct? Or can s be unsampled? It

would also be helpful to unpack the post-event expected genetic distance matrix

equation a little, although this could be in the supplement.

Line 532, what is the separate equation?

line 541, does it scale the units to expected distances or doesn't it?

The "should" qualifier is odd here.

Line 584, missing section number

Regarding future directions, could one imagine a version of FEEMSmix where edges

are removed from the baseline FEEMS fit? For example, imagine a set of

populations separated by unsuitable habitat such that all gene flow between

them occurs by occasional long-range dispersal events across an unsuitable

matrix. In that case, the FEEMS assumption of migration through locally

interacting demes spanning the unsuitable habitat is incorrect, and a LRE

(or set of LRE) inferred by FEEMSmix could replace the local edges in the

baseline FEEMS fit. This is just speculation on my part, I'm not familiar

enough with FEEMS to know if this is feasible.

Reviewer #2: I’m a fan of the EEMS framework. It is used a lot by empirical researchers. Thus, it’s nice to see it built on in productive ways. However, given the apparent similarity to spacemix it feels like the extension done here could have advanced things more substantially.

The authors acknowledge:

“This framework follows closely from existing methods that model the residual from an existing fit as a specialized admixture component (TreeMix,Pickrell & Pritchard (2012); MixMapper, Lipson et al. (2013); SpaceMix, Bradburd et al. (2016)).”

While treemix and mixmapper are conceptually similar, the spatial admixture framework seems to essentially be the spacemix framework. I’m excited to see these ideas implemented to EEMS, and there’s lots of advantages to the EEMS framework (in terms of speed and interpretability). However, I think the authors should explain in a few sentences the admixture model that was implemented in Spacemix in their introduction, and where their spatial admixture model differs, as few readers will reread the earlier references.

Given the similarity to previous methods, it is slight shame that the new method didn’t go further in their model of instantaneous admixture. For example, if I’m understanding it correctly, only a single tile on the grid receives the pulse of admixture, but the authors could’ve allowed some small number of adjacent tiles to share the source admixture (this is briefly mentioned in the discussion). One suggestion is that the authors could simulate under this model and show how this shows up in the model. For example, in the human data analysis there look to be quite a few arrows highlighting similar admixture routes, which could be due to this limitation of the model.

It's notable in the human data analysis that the improvement in R2 with number of admixture edges (Figure 5e) keeps almost flatlining and then making large jumps. Does this suggest that the procedure for choosing the next arrow to add is potentially sub-optimal? If I am understanding correctly, the authors choose to prioritize fitting admixture arrows to demes that appear most often in their outlier pairs (~line 175). Does this prioritize adding arrows to regions where there are many sample locations affected, which is good, but not take into account the magnitude of the outliers? I wonder if using a different metric to choose the arrows would help, e.g. that includes the number and magnitude of the outliers.

Reviewer #3: This manuscript presents an extension of the EEMS method that reconstructs spatial dispersal dynamics from the analysis of geo-referenced genetic data. More specifically, the new technique accommodates for long-range, recent migration events between demes besides "standard" IBD patterns. Simulations along with the analysis of two real data sets were performed in order to illustrate the relevance of the proposed approach.

Reconstructing past dispersal/migration events from the analysis of genetic data is a central endeavour in evolutionary biology and ecology. It is indeed paramount to our understanding of the forces shaping the spatial dynamics of related lineages during the course of their evolution. The EEMS framework provides graphical summaries that are straightforward to interpret, making that tool most useful to evolutionary biologists in practice. Relaxing the strict IBD assumption whereby long-range migrations are now authorised is a substantial improvement. Yet, assessing the relevance of the new approach requires further investigation in my opinion. I list below some points that may provide some guidance in that respect.

### Simulations

Current simulations do not provide enough evidence about the limitations of the proposed methods. In particular, FEEMSmix relies on the hypothesis of recent long-range migration of individuals from destination to source demes. The manuscript does not provide any specifics about the actual meaning of "recent" here. It is unfortunate as the power with which FEEMSmix detects long-range events probably depends on the age of those events.

Sampling considerations also probably matter. Indeed, data may no longer convey signal about the directionality of migration in situations where the recipient deme is isolated (e.g. ((A,B),(A,A,A,A)) suggests a recent migration from region A to B or a migration from B to A. Both scenarios are equally parsimonious, whereas ((B,B,A,B),(A,A,A,A)) is clearly indicative of a B to A migration).

On a related matter, dense and spare sampling are conducted in a uniform manner on the grid as far as I could understand. In practice, sampling is driven by practical considerations (accessibility, funding available, etc.), generally making it highly heterogeneous, i.e. non-uniform. Testing the robustness of the proposed technique to various (non-uniform) sampling patterns seems important if this method is to be applied to a broad variety of data sets.

If the authors see it fit, exploring the behavior of FEEMSmix under a panmictic model would also be a nice addition, illustrating the benefits of using this approach for testing the null hypothesis of no correlation between genetic and physical distances.

### Likelihood of 'FEEMSmix' model

Although the description of the algorithm which FEEMSmix relies on gives a fair overview of the underlying rationale, specific details are difficult to grasp. For instance, in Equation (1), I could not understand the second line (it should read (1-*c*)*T*_*sd*_ + *c**T*_*ss*_, I think). More importantly, the way values of Δ_*ij*_' are calculated is very difficult to understand. In particular, it is not clear what *R*_*is*_, *R*_*sj*_, and \hat{q}_s correspond to (I could not find their definitions elsewhere in the manuscript).

### Miscellaneous comments, questions and remarks

* Line 161: "single single"

* Lines 165-172: What is the rationale for using a threshold of 2 log-likelihood units for testing the directionality of long-range migration events? Is it related to a quantile of a Chi-square distribution? Should that threshold vary as a function of the sample size?

* Lines 310-318: The leave-one-out cross-validation procedure conducted here is quite interesting but the obtained results are described too briefly, in a vague manner unfortunately.

* Figure 5: the difference of R^2 with and without LRE seems very small here (compared to that observed for the simulations). Is there any way one could formally test that adding LREs leads to a statistically significant increase of the log-likelihood?

* Line 556: remove "not"?

* Lines 575-578: what is the motivation for reformulating the expected covariance matrix the way that the authors did?

* Line 584: Section,

-Stéphane Guindon-

**Have all data underlying the figures and results presented in the manuscript been provided?**

Reviewer #1: Yes

Reviewer #2: Yes

Reviewer #3: Yes

PLOS authors have the option to publish the peer review history of their article (what does this mean?). If published, this will include your full peer review and any attached files.

Reviewer #1: No

Reviewer #2: No

Reviewer #3: **Yes: **Stéphane Guindon

**Figure resubmission:**
---

## [Decision Letter · Decision Letter 1]

2 Sep 2025

Dear Dr Shastry,

We are pleased to inform you that your manuscript entitled "Jointly representing long-range genetic similarity and spatially heterogeneous isolation-by-distance" has been editorially accepted for publication in PLOS Genetics. Congratulations!  There are few minor things to fix before your final version submission. 

Yours sincerely,

Gideon S. Bradburd

Guest Editor

PLOS Genetics

Kelly Dyer

Section Editor

PLOS Genetics

Aimée Dudley

Editor-in-Chief

PLOS Genetics

Anne Goriely

Editor-in-Chief

PLOS Genetics

Reviewer's Responses to Questions

**Comments to the Authors:**

Reviewer #1: I am satisfied by the author response to my original concerns about

the overfitting potential of FEEMSmix. The addition of stopping criteria

based on R2 and LR tests is useful, though I agree with the authors

that a heuristic exploratory approach is probably the best way to go here.

The additional material added to the Discussion about how to interpret LRE

and when LRE might be misleading is also appreciated. I have no further

concerns.

Reviewer #2: I thank the authors for their work, the revisions look good.

Reviewer #3: This revised version addresses the issues raised in the previous round of reviews as far as I am concerned. The authors did a substantial amount of work to better illustrate the strength and limitation of their approach. I am confident that FEEMSMix will be widely used in the landscape/spatial genetics community. I therefore recommend publication of the manuscript.

Minor points:

Line 179. Remove dot after "deme".

Line 182: replace o with d in O(o^2) and O(o)?

Line 194: please define x_ij beforehand.

Line 195: "over all outlier pairs" -> indicate that these pairs are restricted to those that have deme i as one of the two elements in the pair?

Line 258: "issues" -> "issue", "necessitate" -> "necessitates"

Lines 339, 342: remove "if" in "with if" and "and if"

Line 344: "as the are no original lineages"

Line 437: "to assess the robustness"

**Have all data underlying the figures and results presented in the manuscript been provided?**

Reviewer #1: Yes

Reviewer #2: Yes

Reviewer #3: Yes

PLOS authors have the option to publish the peer review history of their article (what does this mean?). If published, this will include your full peer review and any attached files.

Reviewer #1: No

Reviewer #2: No

Reviewer #3: No

**Data Deposition**

http://datadryad.org/submit?journalID=pgenetics&manu=PGENETICS-D-25-00146R1

**Press Queries**

---

## [Editor Report · Acceptance letter]

PGENETICS-D-25-00146R1

Jointly representing long-range genetic similarity and spatially heterogeneous isolation-by-distance

Dear Dr Shastry,

We are pleased to inform you that your manuscript entitled " 

Jointly representing long-range genetic similarity and spatially heterogeneous isolation-by-distance" has been formally accepted for publication in PLOS Genetics! Your manuscript is now with our production department and you will be notified of the publication date in due course.

With kind regards,

Anita Estes

PLOS Genetics

On behalf of:
